# Predicting the Susceptibility of Examples to Catastrophic Forgetting

**Guy Hacohen** [1]   **Tinne Tuytelaars** [1]

## Abstract

Catastrophic forgetting – the tendency of neural networks to forget previously learned data when learning new information – remains a central challenge in continual learning. In this work, we adopt a behavioral approach, observing a connection between learning speed and forgetting: examples learned more quickly are less prone to forgetting. Focusing on replay-based continual learning, we show that the composition of the replay buffer – specifically, whether it contains quickly or slowly learned examples – has a significant effect on forgetting. Motivated by this insight, we introduce Speed-Based Sampling (SBS), a simple yet general strategy that selects replay examples based on their learning speed. SBS integrates easily into existing buffer-based methods and improves performance across a wide range of competitive continual learning benchmarks, advancing state-of-the-art results. Our findings underscore the value of accounting for the forgetting dynamics when designing continual learning algorithms.

## 1. Introduction

Recent years have seen a surge in the practical applications of deep learning, yet our mathematical and theoretical understanding of these models remains limited. While many advances are supported by some theoretical insights, the mechanisms behind key phenomena often remain elusive.

One such phenomenon is catastrophic forgetting (French, 1999; Kemker et al., 2018; McCloskey & Cohen, 1989), a core challenge in continual learning (CL) (see reviews: De Lange et al., 2021; Hadsell et al., 2020; Parisi et al., 2019). When models are exposed to new data, their performance on previously learned tasks often degrades. Despite extensive theoretical efforts, a full mathematical understanding of this behavior is still lacking.

In this work, we adopt a behavioral perspective, inspired by psychology and neuroscience, where internal mechanisms are often inaccessible. We train deep networks in various CL settings, observe their forgetting behavior, and infer insights from the resulting patterns.

Our key observation is a last-in-first-out forgetting pattern: examples learned later are more prone to forgetting, while earlier-learned ones are preserved. This aligns with the simplicity bias of neural networks (Shah et al., 2020; Szegedy et al., 2014), where simpler examples are learned first. As a result, simple examples are consistently remembered, while more complex ones are forgotten as new data arrives. This pattern holds across a wide range of architectures, datasets, and training configurations – including variations in learning rates, optimizers, schedulers, epochs, and regularization strategies (see §2, App. D). Fig. 1 visualizes remembered vs. forgotten examples in CIFAR-100.

We further explore how this effect changes as the model's ability to remember examples improves. This improvement is achieved by either expanding the buffer size (§2.4), increasing the network size (App.C.3), or reducing dataset complexity (Fig.11). As memory improves, even slower-learned examples are remembered, leaving only the most complex ones susceptible to forgetting. Nonetheless, the core principle remains consistent: the slower an example is learned, the more likely it is to be forgotten.

Next, we examine how selective replay affects catastrophic forgetting. We rank examples based on their learning speed, and construct different buffer compositions by focusing on examples from a specific range of speeds. We find that selecting moderately fast-learning examples – the simplest and quickest to learn, which are still complex enough to benefit from replay – is particularly effective. As the model's ability to remember improves, so does the optimal buffer focus shift progressively toward slower-to-learn examples.

Most competitive buffer-based methods rely on random sampling to fill the replay buffer, treating all examples from the old task equally. To demonstrate the potential of accounting for the susceptibility of examples to forgetting, we introduce a simple sampling strategy for replay buffers called speed-based sampling (SBS). SBS tracks the speed at which

[1]ESAT-PSI, KU Leuven, Belgium. Correspondence to: Guy Hacohen <guy.hacohen@mail.huji.ac.il>, Tinne Tuytelaars <Tinne.Tuytelaars@esat.kuleuven.be>.

*Proceedings of the 42ⁿᵈ International Conference on Machine Learning*, Vancouver, Canada. PMLR 267, 2025. Copyright 2025 by the author(s).

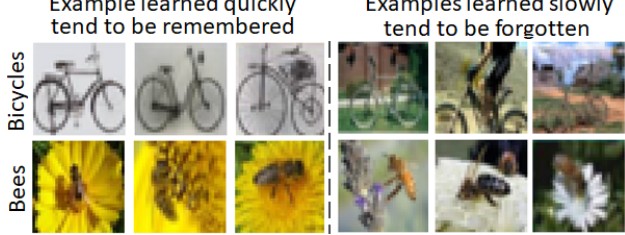

Figure 1: Test examples that networks tend to either remember or forget. Networks were trained on 2 tasks from CIFAR-100. We show the examples that were learned quickest (left) and slowest (right). Slowest examples were often forgotten, while the quickest were rarely forgotten.

each example is learned during the old task training and prioritizes sampling the quickest-to-learn examples that are still prone to forgetting (details in §3.1). This approach is computationally and memory-efficient, requiring just one float per example computed during the forward pass, and integrates easily with any replay-based CL method. Despite its simplicity, SBS consistently improves performance across datasets and buffer sizes, demonstrating the practical value of our insights.

### 1.1. Our Contribution

- Observing a connection between an example's learning speed and its susceptibility to catastrophic forgetting.

- We show that this observed relationship also persists across CL settings, even as models become more capable of remembering.

- Demonstrating that this phenomenon is robust across diverse datasets and hyperparameters.

- Showing that restricting the replay buffer to examples of specific learning speeds significantly affects forgetting across different continual learning scenarios.

- Proposing a practical application of the observed phenomena, by introducing a novel sampling strategy for replay buffers, called speed-based sampling (SBS). SBS integrates with existing continual learning algorithms, significantly improving their performance across a wide range of methods and advancing the state of the art.

### 1.2. Related Work

**Catastrophic forgetting.** Most existing work on catastrophic forgetting focuses on mitigation (Kirkpatrick et al., 2017; Lee et al., 2017; Li et al., 2019; Ritter et al., 2018; Serra et al., 2018; Verwimp et al., 2025), rather than its characterization. Some works have explored catastrophic

forgetting from a model-centric perspective: Nguyen et al. (2020); Ramasesh et al. (2021) showed differences in forgetting patterns across model layers, while Mirzadeh et al. (2020); Pfülb & Gepperth (2019) demonstrated the impact of training hyper-parameters. Nguyen et al. (2019) identified task complexity as a factor influencing forgetting rates. In contrast, our approach is data-centric and behavioral – we link forgetting to the speed at which individual examples are learned

**Simplicity Bias and Selective replay.** Previous studies have shown that neural networks tend to learn simple patterns before more complex ones (Cao et al., 2021; Gissin et al., 2020; Gunasekar et al., 2018; Heckel & Soltanolkotabi, 2020; Hu et al., 2020; Jin & Montúfar, 2023; Kalimeris et al., 2019; Pérez et al., 2019; Soudry et al., 2018; Ulyanov et al., 2018) – a phenomenon known as simplicity bias (Dingle et al., 2018; Shah et al., 2020). Our work extends this concept to catastrophic forgetting, revealing a reverse simplicity bias – complex examples are more likely to be forgotten than simple ones.

Our method leverages this insight by selectively replaying examples based on learning speed, indirectly controlling for complexity. Related paradigms such as curriculum learning (Bengio et al., 2009; Hacohen & Weinshall, 2019) and self-paced learning (Kumar et al., 2010), structure training around example complexity, demonstrating that such structure can improve generalization. However, unlike these approaches, we operate in a continual learning setting. We do not modify the original data or training sequence. Instead, we influence learning solely through selective replay, using example complexity as the criterion and observing its effect on catastrophic forgetting.

Our work also aligns with insights from semi-supervised learning, transfer learning, and active learning, where the complexity of the limited labeled data significantly affects algorithm design and performance (Hacohen et al., 2022; Yehuda et al., 2022; Hacohen & Weinshall, 2023a;b; Fluss et al., 2023; Chen et al., 2024). In all these settings, understanding and leveraging the distribution of example complexity can be key to effective learning.

**Forgetting Dynamics.** Several studies investigate forgetting dynamics during training. Maini et al. (2022); Toneva et al. (2019) show that, in non-continual learning settings, certain examples are inherently less prone to forgetting during training. Millunzi et al. (2023) highlighted distinct forgetting behaviors for noisy versus clean labels, proposing adjustments to the rehearsal process to account for these differences. Our findings build on this body of work by further identifying which examples are most susceptible to forgetting, linking catastrophic forgetting to simplicity bias. We propose a method to proactively identify examples susceptible to forgetting and demonstrate how these insights

can be applied to optimize replay buffer sampling across various CL methods.

**Replay buffer sampling functions.** Replay-based methods mitigate forgetting by storing past examples in a fixed-size buffer. While numerous sampling strategies have been proposed (Aljundi et al., 2019; Benkő, 2024; Buzzega et al., 2021; Rebuffi et al., 2017; Tiwari et al., 2022; Wiewel & Yang, 2021), many are limited to specific conditions or methods. As a result, uniform sampling remains common (Buzzega et al., 2020; Guo et al., 2020; Kirkpatrick et al., 2017; Lopez-Paz & Ranzato, 2017; Prabhu et al., 2020; Ramesh & Chaudhari, 2022; Rolnick et al., 2019). In contrast, our proposed SBS offers a lightweight, general approach that consistently enhances replay-based methods across tasks, datasets, and buffer sizes.

## 2. Catastrophic Forgetting Vs. Learning Speed

In this section, we observe the behavior of neural models when they exhibit catastrophic forgetting under different settings, drawing a connection between the speed at which a model learns examples, and its likelihood to forget them.

### 2.1. Definitions

We examine the CL classification setting, where the training dataset $\mathcal{D} = \{\mathcal{X}, \mathcal{Y}\}$ consists of examples $x \in \mathcal{X}$ and their corresponding labels $y \in \mathcal{Y}$. This dataset is divided into $T > 1$ tasks, with each task $t \in [T]$ represented as $\mathcal{D}_t = \{\mathcal{X}_t, \mathcal{Y}_t\}$, and $\mathcal{D} = \bigcup_{t=1}^{T} D_t$. There is no overlap between data of different tasks. A deep model $f : \mathcal{X} \to \mathcal{Y}$ is sequentially trained on each task $\mathcal{D}_t$ from $t = 1$ to $t = T$, with $E \in \mathbb{N}$ epochs per task. Each task also has a separate test set from the same distribution. During training on a task $\mathcal{D}_t$, the model $f$ is provided with a replay buffer $B$ of fixed-memory size, containing examples from previous tasks $1, ..., t-1$. Notably, $|B| \ll |\mathcal{D}|$, meaning only a fraction of the data can be stored in the replay buffer. Note that the buffer may be empty ($B = \emptyset$), representing training $f$ without a replay buffer.

**Learning speed.** Neural networks demonstrate a trend in learning examples, where certain examples are consistently learned before others across various deep neural models (Baldock et al., 2021; Choshen et al., 2022; Hacohen et al., 2020). This orderliness is often connected to the simplicity bias phenomenon (Shah et al., 2020; Hacohen & Weinshall, 2022), as examples learned earlier are generally simpler than those learned later. Here, we measure the speed at which an example is learned, which also serves as a proxy for the example's complexity. Formally, given a model $f$ and an example $(x, y)$ from either the training or test data, we define the *learning speed* of the example $(x, y)$ as:

$$learning\_speed(x, y) = \frac{1}{E} \sum_{e=1}^{E} \mathbb{1}[f_t^e(x) = y] \quad (1)$$

Here, $f_t^e$ denotes the intermediate model $f$ after training for $e \in [E]$ epochs on task $\mathcal{D}_t$. This definition is based on the *accessibility score* introduced by Hacohen et al. (2020), which we adapt for computation using a single model instead of an ensemble, facilitating efficient evaluation throughout the training of $f$. Intuitively, this score correlates with how quickly the model learns an example: if an example is correctly classified from the early stages of learning, it will sustain $f_t^e(x) = y$ for more epochs, resulting in a higher *learning speed*. A discussion of alternative metrics to learning speed can be found in App. G. A detailed explanation of the relationship between the example's learning speed and this score is provided in Hacohen et al. (2020), and thus is not repeated here.

**Computing learning speed.** To calculate the learning speed of each example, we maintain a boolean matrix, called the epoch-wise classification matrix, $M \in \{0, 1\}^{E \times |\mathcal{D}_t|}$ throughout the training on task $\mathcal{D}_t$, indicating whether the model correctly classified each example during learning. Specifically, for example $(x_i, y_i) \in \mathcal{D}_t$, we have $M_{e,i} = \mathbb{1}[f_t^e(x_i) = y_i]$. An example's learning speed is the average of the $M$ across epochs. This matrix can be computed during the forward pass of the network, incurring minimal computation overhead. Furthermore, memory usage can be reduced by directly computing the matrix's mean during training, maintaining a single float for each example.

### 2.2. Preliminaries

**Class Incremental vs. Task Incremental Learning.** Although similar, task incremental learning differs (TIL) from class incremental learning (CIL) by providing the task identity for each example during testing, allowing the use of the specific classifier for the classes within the task. We conducted experiments for both TIL and CIL. Our qualitative results across all experiments were consistent for both TIL and CIL. Therefore, to avoid redundancy, we present in the main paper results only for TIL setting, and repeat all the figures and experiments for CIL in App. B.

**Datasets.** We investigated various image continual learning classification tasks using split versions of several image datasets, including CIFAR-10, CIFAR-100 (Krizhevsky et al., 2009), and TinyImageNet (Le & Yang, 2015). The data is split into $T$ tasks by partitioning the classes into $T$ equal-sized subsets. This partitioning is denoted as dataset-T. For example, splitting CIFAR-10 into 5 classes is denoted as CIFAR-10-5, comprising 5 tasks, each with 2 distinct classes. Unless otherwise specified, classes are divided into

tasks according to the original order they appeared in the dataset (often alphabetically).

For detailed implementation details, including the architectures and hyper-parameters used, please see App. A.

### 2.3. Learning Speed vs. Catastrophic Forgetting

We begin by exploring catastrophic forgetting in a simplistic case, without using replay buffer ($B = \emptyset$), and with only $T = 2$ tasks. In Fig. 2a, we plot the mean test accuracy of each task when training 10 networks on CIFAR-100-2. The dashed black line marks the transition from the first to the second task. Consistent with prior research, the networks exhibit significant catastrophic forgetting, evidenced by a sharp decline in the accuracy of the first task during training on the second task. Our objective is to characterize those examples where the network successfully classified after the first task but failed after the second task.

To simultaneously track learning speed and forgetting, we maintain the epoch-wise boolean classification matrix $M$ for each network (formally defined in §2.2). Throughout the training of both tasks, we record, for each epoch, whether the network correctly classified each example from the first task. The initial 100 epochs represent classification during the first task, enabling us to observe the learning speed of the different examples, while the subsequent 100 epochs allow us to monitor the forgetting of the first task during the training of the second task.

In Figs. 2(b-c), we examine the epoch-wise classification matrices for the train and test sets, respectively. To aid visualization, instead of plotting the epoch-wise classification matrix for each network, we aggregate the networks from all runs using majority voting. The examples in the matrix are sorted by their learning speed. We observe a correlation between learning speed and catastrophic forgetting: examples learned more quickly during the first task tend to remain correctly classified throughout the second task, whereas slower-learned examples are prone to immediate misclassification upon task transition, making them susceptible to forgetting. Essentially, slower learning speeds of examples correlate with a higher probability of forgetting by deep models. Additional classification matrices, for different datasets and different amounts of tasks, can be found in App. C and Fig. 15.

To quantify this relationship, we define an example in the first task as "remembered" by the network if it is classified correctly at the conclusion of both the first and second tasks. In Fig. 2d, we plot, for each example in the test set, the % of networks that remembered it vs. its mean learning speed across all 10 networks. For visualization, we group examples remembered by a similar percentage of networks and average their mean learning speeds. We observe a strong cor-

relation ($r = 0.995, p \leq 10^{-10}$), indicating that networks are more likely to remember quickly learned examples.

While the results presented here pertain to a simplified case, this phenomenon is robust across diverse datasets, architectures, and hyperparameters. For additional experiments, please refer to App. C, D.

### 2.4. Adding a Replay Buffer of Different Sizes

To evaluate replay-based methods, we incorporate a replay buffer in our experiments. Similarly to §2.3, we trained 10 networks on CIFAR-100-2, with replay buffers of varying sizes ($1k$, $3k$, $10k$). In Figs. 3(a-c), we plot the epoch-wise classification matrix of the test data for each case. The examples appear in each matrix in the order of their learning speed, from slow (bottom) to quick (top). As expected, integrating a replay buffer mitigates some catastrophic forgetting, boosting the model performance on the first task, with larger buffers yielding better results. Nevertheless, similar to the scenario without a replay buffer $B = \emptyset$, a relationship exists between learning speed and catastrophic forgetting: networks tend to remember fast-learned examples while forgetting those learned later even when using a replay buffer. Notably, models with larger buffer sizes remember most of the examples that models with smaller buffer sizes do, while additionally remembering gradually slower-learned examples. Similar results occur when changing the network architecture, and when decreasing the dataset complexity (see App. C, D).

We quantify the relationship between the buffer size and remembered examples' learning speed. Training 10 networks on CIFAR-100-2 with various buffer sizes, we plot the mean learning speed of the remembered examples for each buffer size. A strong correlation emerges ($r = 0.966, p \leq 10^{-6}$): models with smaller buffer sizes remember quickly-learned examples, whereas larger models with larger buffers enable the model also to remember slower-learned. These results are plotted in Fig. 3d. Similar results were achieved across diverse datasets, architectures, hyperparameters, and continual learning algorithms, see App. C, D.

### 2.5. Different Buffer Compositions and Sizes

Replay-based continual learning methods often sample the replay buffer uniformly, treating quickly and slowly learned examples equally. Here, we analyze different replay buffer compositions, focused on examples learned at specific speeds. To focus the buffer on examples learned at a certain speed, we set 2 threshold, *quick* and *slow*, and sample the buffer uniformly from all the examples that were learned slower than the *quick* threshold and faster and the *slow* threshold. This allows us to bias the replay buffer toward examples learned at certain speeds.

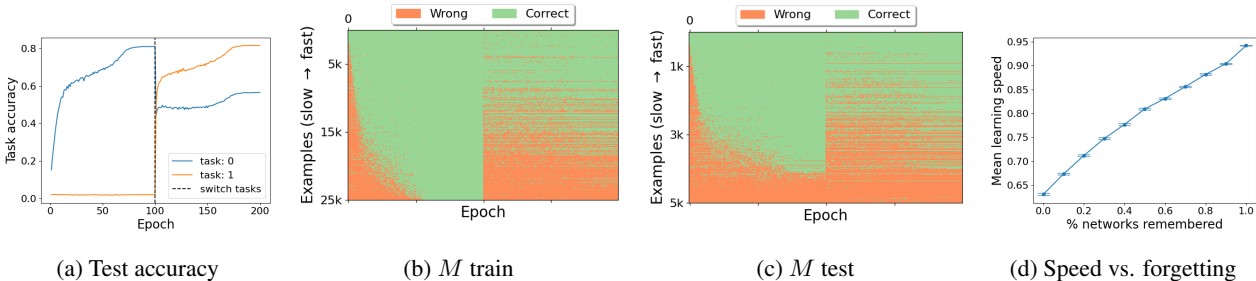

| (a) Test accuracy | (b) $M$ train | (c) $M$ test | (d) Speed vs. forgetting |

Figure 2: Forgetting as a function of the learning speed. We trained 10 networks on CIFAR-100-2, without a replay buffer. (a) Mean test accuracy of each task, where the dashed line marks the task switch. The models forget much of the first task after the switch. (b-c), the first task's binary epoch-wise classification matrices $M$ for train and test data. The y-axis corresponds to different examples, and the x-axis corresponds to different epochs, showing if the examples were classified correctly in each epoch. The order in which the examples are plotted is sorted by the examples' learning speed. Faster-learned examples from the first task are less likely to be forgotten at the end of the second task. (d) The mean learning speed of examples in the first task vs. the % of networks that remember them at the end of the second task. Networks forget more examples learned slowly.

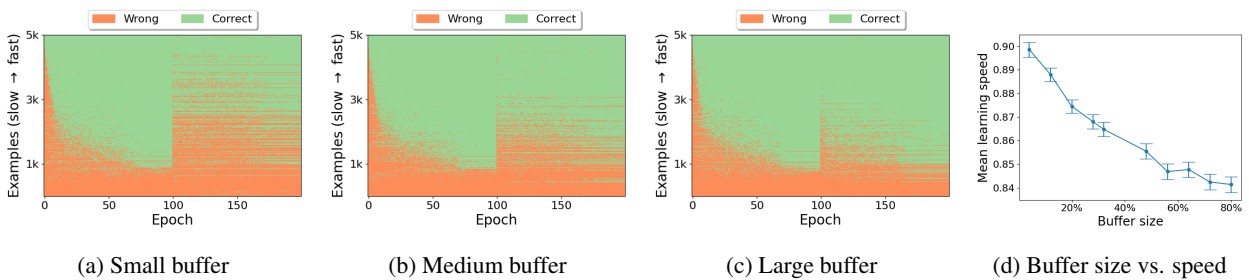

| (a) Small buffer | (b) Medium buffer | (c) Large buffer | (d) Buffer size vs. speed |

Figure 3: Impact of the buffer size on forgetting dynamics. (a-c) Similarly to Fig. 2c, the first task's binary matrices $M$, when adding replay buffer of varying buffer sizes: small ($1k$), medium ($3k$), and large ($12k$). The y-axes denote examples, and the x-axes denote epochs, indicating if the example has been classified correctly by the model at that epoch. The order in which examples appear is sorted by the examples' learning speed. In all cases, examples that were learned faster are more likely to be remembered. With bigger buffers, the networks can remember gradually slower-to-learn examples. (d) The mean learning speed of remembered examples of models with different buffer sizes. Models with bigger replay buffers remember slower-to-learn examples.

We compare replay buffer compositions for CIFAR-100-2 across different buffer sizes. Figs.4(a-b) show the mean final accuracy difference relative to uniform sampling, averaged over 10 runs. Compositions outperforming uniform sampling are in red, while those reducing performance are in blue. The black-boxed point $(quick, slow) = (0, 0)$ represents the uniform baseline. Standard errors are plotted separately (App.E). In both cases, a wide range of buffer compositions (marked in red) significantly improves the final accuracy, suggesting that uniform sampling of the replay buffer is suboptimal.

While many buffer compositions are beneficial for both cases, we see that when the buffer size is smaller, it is more beneficial to remove slowly-learned examples, focusing mainly on the easier examples learned quickly by the models. In contrast, with larger buffers, the optimal focus

in the buffer shifts towards harder, slowly-learned examples, where quickly-learned examples are removed.

In Fig. 4c, we replicate this experiment on TinyImageNet-2, with a large buffer of size $20k$. Similar to CIFAR-100-2, focusing on examples learned midway through learning yields the best performances. Notably, across all cases, the range of thresholds that enhance performance is broad and continuous, indicating that improvements are not limited to specific hyperparameter sets. Similar results are achieved in the multi-task case, see App. C.5.

To assess the robustness of these results, we repeat the experiment from Fig. 4 under various learning settings, including changes to optimizers, architectures, learning rates, training durations, fine-tuning parameters, regularization techniques, and data splits. Across all cases, we observe consistent qual-

itative results. These experiments can be found in App. D.

## 2.6. How Subsequent Tasks Affect Buffer Composition

We empirically examine how subsequent tasks impact the ideal composition of a replay buffer for a given task. We find that regardless of the similarity or dissimilarity between subsequent tasks and the original task, the optimal replay buffer composition remains largely independent and consistent. We conduct experiments with three distinct tasks denoted A, B, and C. By training the model on task A and subsequently introducing either task B or task C while keeping the original task unchanged, we analyze the ideal replay buffer composition under varying subsequent tasks.

We initially focus on tasks from the same datasets. We pick task A to be the first 25 classes of CIFAR-100, task B to be classes 26 to 50, and task C to be classes 51 to 75. In Fig. 5, we train various replay buffer compositions when either task B or task C is after task A. We plot the difference between the mean final accuracy of each buffer composition to a uniformly sampled buffer. Each composition was repeated 10 times. The consistent qualitative results indicate that good buffer compositions for one case are also good for the other, and vice versa. These results suggest that the ideal replay buffer composition of task A is robust to differences between tasks B and C in this context.

Further, we explore scenarios involving either changing the classification task or replacing it with memorization by training on random labels. We adopt the RotNet approach (Gidaris et al., 2018) to change the classification task. We pick tasks A and B to be different subsets of CIFAR-100, while task C involves a rotation classification task, where each example in task A is randomly rotated between $90°, 180°,$ or $270°$ degrees, with the task being to classify these rotations regardless of the original label. For the random case, tasks A and B remain unchanged, while task C involves the same subset of task B, but with labels chosen uniformly at random, as suggested in Zhang et al. (2021). In both scenarios, we observe consistent results: the ideal replay buffer composition of task A remains robust to differences between tasks B and C in both contexts. Further details and heatmaps for the results are available in App. F.

## 3. Speed-Based Sampling (SBS)

Above, we observed a distinctive behavior of neural networks trained in a continual learning setting: when exposed to new data, networks tend to forget previously learned data, with examples that were learned more quickly being the least likely to be forgotten. This section demonstrates that this behavior has practical implications and can be leveraged to mitigate catastrophic forgetting in CL.

We propose Speed-Based Sampling (SBS), a novel sampling strategy for replay buffers that replaces traditional random sampling which is common in various CL methods. SBS prioritizes examples with specific learning speeds, allowing the replay buffer to account for the susceptibility of examples to forgetting. This allows the underlying CL algorithm to focus on the most relevant examples, effectively reducing catastrophic forgetting.

### 3.1. Method Definition

Speed-Based Sampling (SBS) is a sampling strategy for the replay buffers in buffer-based CL at the end of each task. It is compatible with any buffer-based CL algorithm.

During training on task $t$, SBS calculates the classification matrix (defined in §2.1) of task $t$, computing at the end of the task the *learning speed* of each example in $\mathcal{D}_t$. Once training on task $t$ is completed, SBS creates a filtered dataset $\mathcal{D}'_t$ by removing the $q$ quickest-learned examples and $s$ slowest-learned examples from $\mathcal{D}_t$. Examples for the replay buffer are then sampled uniformly from $\mathcal{D}'_t$.

The hyperparameters $q$ and $s$ can be adjusted based on the user's requirements (see below). In scenarios where the buffer size is limited and examples need to be removed (e.g., when handling multiple tasks), examples from previous tasks can be removed from the buffer at random.

Pseudocode for integrating SBS with any buffer-based CL method is provided in Algorithm 1.

**Choosing $q$ and $s$.** The choice of $q$ and $s$ is crucial for the performance of SBS. In our experiments, we determined these hyperparameters by training a RotNet auxiliary task on the same dataset, without using additional data points. Specifically, after training the CL method on $\mathcal{D}_t$, we created a rotated version of the dataset by randomly rotating each image by $0°, 90°, 180°,$ or $270°$. Then we trained the network to classify the rotation of each example. A coarse grid search was performed over $q$ and $s$ values, selecting the ones that yielded the best performance on this task.

While effective in practice, this approach requires training the model multiple times, which can be computationally expensive for larger datasets. However, the smoothness of the parameter space (see Fig.4) enables a coarse grid search to be sufficient. Moreover, since the auxiliary task is relatively simple, each training run is significantly faster than training the model from scratch on the full dataset $\mathcal{D}_t$, as fewer epochs are needed. In addition, the method is robust to hyperparameter choices: a broad range of values for $q$ and $s$ tend to yield improvements, making heuristic settings often adequate. For example, setting $q = s = 20\%$ consistently enhances performance across all evaluated datasets and generalizes well to unseen ones. This is evident in Figs. 4(a–c), 9(a–b), 16(a–b), 17(a–b), 18(a–c), 19(a–c), 20(a–c), 21(a–c), and 24(a–c), where the point correspond-

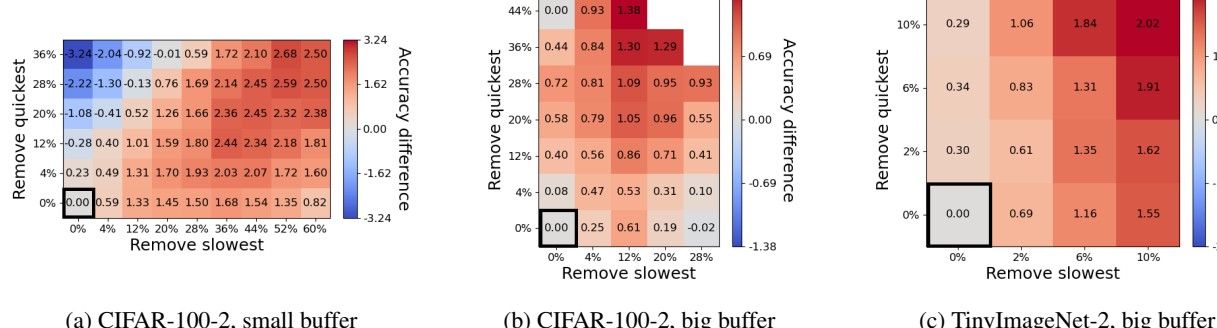

(a) CIFAR-100-2, small buffer      (b) CIFAR-100-2, big buffer      (c) TinyImageNet-2, big buffer

Figure 4: Comparison of buffer compositions for various buffer sizes, removing slowly and quickly learned examples (see §3.1). The mean final accuracy difference between each composition and a randomly sampled buffer is shown, averaged over 10 runs. Red shades indicate improved performance over uniform sampling, while blue indicates worse performance. The baseline (uniformly sampled buffer) appears at $(0\%, 0\%)$, marked with a black box. (a) CIFAR-100-2 with a small $1k$ buffer: removing slower-to-learn examples is more beneficial. (b) CIFAR-100-2 with a large $10k$ buffer: removing faster-to-learn examples is more beneficial. (c) TinyImageNet-2 with a $20k$ buffer. Across all scenarios, diverse buffer compositions are beneficial, with smaller buffers benefiting more from faster-to-learn examples.

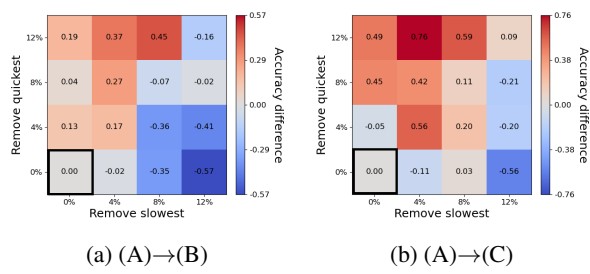

(a) (A)→(B)      (b) (A)→(C)

Figure 5: Buffer compositions when training on the same initial task, followed by different subsequent tasks. We use three subsets of CIFAR-100: A, B, and C (see §2.6). (a) Task A followed by B. (b) A followed by C. Despite the differences in subsequent tasks, the buffer compositions for the original task similarly mitigate catastrophic forgetting.

---

**Algorithm 1** Training $CL$ method with SBS.

**Input:** $\mathcal{D}_t, |B|, E$, amount of quick/slow to remove $q, s$.
**Output:** buffer of size $|B|$
     Classification_Matrix $\leftarrow 0^{E \times |\mathcal{D}_t|}$
     **for** $e = 1, ..., E$ **do**
         Train the model $f$ one epoch using the $CL$ method
         **for** $(x_i, y_i) \in \mathcal{D}_t$ **do**
             Classification_Matrix$[e, i] \leftarrow \mathbb{1}[f(x_i) = y_i]$
         **end for**
     **end for**
     $Speed \leftarrow$ Mean(Classification_Matrix, axis=0)
     $\mathcal{D}'_t \leftarrow$ remove $q\%$ quickest and $s\%$ slowest from $\mathcal{D}_t$
     **return** $|B|$ examples sampled uniformly from $\mathcal{D}'_t$

---

ing to $q = s = 20\%$ consistently significantly outperforms random sampling, even if it does not always achieve the optimal result. Table 1 further compares the performance of SBS with and without hyperparameter tuning. Even in the fixed setting (SBS-fix, with $q = s = 20\%$), the method consistently improves performance across all datasets and buffer sizes tested.

Further refinements can also be guided by task-specific rules. Based on the results of §2.5, if the task is relatively easy for the model (resulting in less catastrophic forgetting), it is beneficial to focus more on slower-to-learn examples, increasing $q$ and decreasing $s$. Conversely, for more challenging tasks where forgetting is pronounced, decreasing $q$ and increasing $s$ is more appropriate.

**Running time.** Given a set of $q$ and $s$ parameters, SBS

introduces minimal computational overhead to the underlying CL method. The classification matrix required by SBS is computed using the classifications of each example during each epoch, which are already part of the forward pass. While the matrix has a size of $E \times |\mathcal{D}_t|$, potentially large for many epochs, this memory cost can be reduced by maintaining only a running mean for each example, lowering the storage requirement to $|\mathcal{D}_t|$. Additionally, as SBS samples examples randomly from a smaller data pool, the sampling process has a similar runtime to standard random sampling.

### 3.2. SBS Improves Various CL Algorithms

Many replay-based CL algorithms rely on uniform sampling for their replay buffers. Here, we evaluate the impact of replacing uniform sampling with SBS across such algorithms,

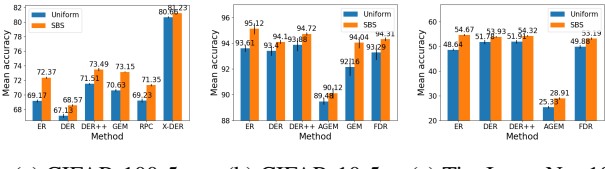

(a) CIFAR-100-5     (b) CIFAR-10-5     (c) TinyImageNet-10

Figure 6: SBS vs. uniform sampling with various continual learning methods. Bars show the mean final test accuracy across all tasks, with random sampling in blue, and SBS in orange. Error bars denote the standard error. (a) CIFAR-100-5 (b) TinyImageNet-10, with a buffer of 500 examples. SBS improves performance across all methods and datasets.

including DER, DER++ (Buzzega et al., 2020), X-DER (Boschini et al., 2022), AGEM (Chaudhry et al., 2019), ER (Rolnick et al., 2019), GEM (Lopez-Paz & Ranzato, 2017), RPC (Pernici et al., 2021), and FDR (Titsias et al., 2020). Our results show that SBS consistently improves the performance of all tested continual algorithms, advancing the state of the art in multiple image classification tasks.

We compared the mean final test accuracy across tasks for each method trained with either uniform sampling (as in the original works) or SBS. While not all methods perform equally well in every scenario, SBS significantly improves accuracy compared to random sampling in all cases. We focused on small buffer sizes, as these are the primary use cases addressed in the original works.

Fig. 6 shows results on CIFAR-10-5, CIFAR-100-5, and TinyImageNet-10 with a fixed buffer size of 500. Each task was trained for 50 epochs. As the buffer size is fixed, examples were removed starting from the second task to accommodate new data. For SBS, examples were removed at random, while uniform sampling employed either random or reservoir sampling, following the strategies outlined in each method's original implementation. The $q$ and $s$ hyperparameters for SBS were determined using the RotNet auxiliary task, as detailed in §3.1.

When training with multiple tasks, different $q$ and $s$ hyperparameters could be used for each task. However, to reduce computational cost and for simplicity, we apply the same $q$ and $s$ values across all tasks in multi-task scenarios. While choosing separate $q$ and $s$ for each task would likely improve performance further, we observe that SBS's performance boost is especially pronounced in multi-task settings (see App. C.5), indicating its particular effectiveness in such scenarios.

### 3.3. SBS Vs. Other Replay Buffer Sampling Methods

While numerous replay buffer sampling methods have been proposed over the years, uniform sampling remains preva-

Table 1: Comparison of replay buffer sampling methods. Each entry represents the mean test accuracy across all tasks for 10 networks trained with experience replay using different sampling methods. The highest accuracy in each scenario is in bold. For visualization, standard errors are reported separately in Table 4. While certain methods, such as Herding (small buffers) and GSS (large buffers), perform well in specific cases, they fail to consistently outperform random sampling across all scenarios. In contrast, SBS consistently achieves the highest accuracy across all scenarios.

| Buffer | CIFAR-100-20 | | CIFAR-10-5 | | TinyImageN-2 | |
|---|---|---|---|---|---|---|
| | $1k$ | $10k$ | $1k$ | $10k$ | $1k$ | $10k$ |
| Random | 51.75 | 71.25 | 79.48 | 82.4 | 49.81 | 61.48 |
| Max Ent | 48.3 | 69.83 | 77.28 | 81.72 | 44.49 | 57.88 |
| IPM | 49.91 | 71.7 | 79.3 | 80.57 | 48.58 | 61.97 |
| GSS | 48.13 | 72.9 | 76.71 | 84.13 | 49.36 | 62.89 |
| Herding | 54.0 | 73.51 | 81.8 | 81.03 | 51.17 | 62.44 |
| LARS | 51.82 | 71.41 | 79.17 | 84.93 | 50.22 | 62.78 |
| SBS-fix | 54.65 | 73.72 | 82.24 | 86.15 | **52.15** | 63.12 |
| SBS | **55.43** | **74.80** | **83.59** | **89.17** | **52.15** | **63.16** |

lent in practical implementation within CL algorithms. This preference may stem from the specific settings or algorithmic modifications required for other sampling methods to work well. These methods do not consistently improve different CL algorithms across datasets and buffer sizes. In contrast, SBS improves a wide range of continual learning algorithms across various buffer sizes and datasets.

We explored various replay buffer sampling strategies, including uniform sampling, max entropy, IPM (Zaeemzadeh et al., 2019), GSS (Aljundi et al., 2019), Herding (Rebuffi et al., 2017), and LARS (Buzzega et al., 2021). We trained 10 networks on TinyImageNet-2, CIFAR-100-20, and CIFAR-10-5 datasets with buffer sizes of $1k$ and $10k$ using experience replay. Table 1 presents the mean test accuracy across all tasks for each case. Additional results on CIFAR-10-2 and CIFAR-100-2 can be found in App. C.5.

SBS consistently outperforms all other sampling strategies across all evaluated scenarios. In contrast, alternative methods show effectiveness only in specific settings. For example, Herding – favoring examples whose features are close to the class mean – surpasses uniform sampling at smaller buffer sizes but underperforms with larger ones. This likely reflects its bias toward typical examples, which are learned early and thus more useful when the replay buffer is small. Conversely, GSS, which prioritizes examples with large gradient distances, excels with larger buffers but lags with smaller ones, as its preference for slowly learned examples becomes more beneficial when a larger replay buffer is available. Table 1 also includes SBS-fix, a variant of SBS that uses default hyperparameters ($q = s = 20\%$, as

recommended in §3.1) without fine-tuning. While SBS-fix underperforms SBS, it still consistently outperforms other sampling methods, indicating that extensive hyperparameter tuning is often unnecessary.

### 3.4. Limitations

One limitation of SBS is that the *learning speed* score is based on the number of training epochs. With a small number of epochs, the score becomes overly discrete, which reduces the precision of the evaluation. However, as shown in App. D and Fig. 20, SBS remains effective even with a small number of epochs, as long as the network converges, although with increased noise. Nonetheless, many continual learning scenarios emphasize stream-like settings, which typically involve only a single epoch. These scenarios are not well-supported by SBS, as the *learning speed* fails to adequately capture how different examples are learned over time. Alternative methods for approximating example complexity are needed, which we leave as future work.

### 3.5. Discussion

This paper takes an observational approach to catastrophic forgetting, identifying a link between the speed at which examples are learned and their likelihood of being forgotten – examples learned more quickly are less prone to forgetting when new data is introduced. SBS was proposed as a simple sampling function that demonstrates the practical utility of this observation. By modifying the contents of the replay buffer, SBS encourages algorithms to focus on examples learned at varying speeds. Indeed, integrating SBS into a wide range of continual learning algorithms across diverse settings yields significant improvements, suggesting that this link is an important factor in mitigating catastrophic forgetting. This insight opens avenues for future work, such as designing more sophisticated sampling strategies or incorporating learning speed more directly into continual learning methods.

The simplicity of SBS also brings practical advantages: it is computationally efficient, easy to implement, and compatible with any buffer-based continual learning approach. As continual learning methods evolve, SBS remains relevant due to its ease of integration and broad applicability.

### Impact Statement

This paper presents work whose goal is to advance the field of Machine Learning. There are many potential societal consequences of our work, none of which we feel must be specifically highlighted here.

## Acknowledgment

This work was supported by ERC-2020-AdG grant 101021347, KeepOnLearning.

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

# Appendix

## A. Implementation Details

**Architectures and hyper-parameters.** In our experiments, unless stated otherwise, we trained ResNet-18 for $E = 100$ epochs per task. We employed a base learning rate of $0.1$ with a cosine scheduler, SGD optimizer, momentum of $0.9$, and weight decay of $0.0005$. All networks were trained on NVIDIA TITAN X. These hyperparameters were chosen arbitrarily based on their performance during joint dataset training and were consistent across all experiments. We anticipate consistent qualitative results despite potential variations in these hyperparameters. When introducing a replay buffer, we employed an experience replay strategy (Rolnick et al., 2019), alternating batches of data from the new task and the replay buffer. When integrating with other continual learning algorithms, we evaluated all methods within the framework of (Boschini et al., 2022; Buzzega et al., 2020), using the hyperparameters suggested in the original papers, changing only the replay buffer sampling to SBS, keeping the rest of the method intact.

## B. Class Incremental Learning

There are two primary paradigms in continual learning: class incremental learning (CIL) and task incremental learning (TIL). The distinction lies in how tasks are handled. In TIL, task identities are known during inference, enabling the use of specialized classifiers within each task, whereas in CIL, the model must infer both the task and class identity without prior knowledge, often leading to lower performance. These paradigms address different real-world scenarios, and methods optimized for one sometimes underperform on the other, necessitating separate evaluations in continual learning research.

Our work examines the relationship between the speed at which examples are learned and their susceptibility to forgetting in continual learning. Despite the differences between CIL and TIL, our study reveals that the same qualitative results and the same conclusions can be made in both cases. Therefore, to avoid repetition in the main text, we focus there on TIL only, as it isolates forgetting at the model level rather than conflating it with final-layer interference. To complement this, we reproduce all key experiments under CIL settings in this appendix, highlighting similar conclusions despite the inherent challenges of CIL.

Figs. 7, 8, 9, 10, 17 present CIL counterparts to the main text's results, Figs. 2, 3, 4, 6, 16 respectively. While CIL results exhibit lower overall performance, the trends remain consistent: examples learned faster are less likely to be forgotten, and the correlations between the buffer size and forgetting and the correlation between the learning speed and forgetting are strong. Notably, optimal buffer composition in CIL scenarios (Figs. 9, 17) slightly favors keeping more quickly learned examples compared to TIL, reflecting the added complexity of task inference in CIL.

Fig. 10 reproduces results from Fig. 6, demonstrating the impact SBS sampling strategy across CIL datasets and algorithms. SBS consistently yields significant performance gains, reaffirming its effectiveness across diverse continual learning challenges.

Finally, Table 2 shows the results of the experiments in Table 1 in the CIL settings, showing similar qualitative behavior.

Table 2: The same table as Table 1, when doing class-incremental learning instead of task-incremental. Each entry represents the mean test accuracy across all tasks for 10 networks trained with experience replay using different sampling methods. The highest accuracy in each scenario is in bold.

| | CIFAR-100-20 | | CIFAR-10-5 | | TinyImageN-2 | |
|---|---|---|---|---|---|---|
| Buffer | $1k$ | $10k$ | $1k$ | $10k$ | $1k$ | $10k$ |
| Random | 42.9 | 48.5 | 57.74 | 79.47 | 9.99 | 27.4 |
| Max Ent | 41.76 | 47.82 | 57.11 | 69.13 | 9.47 | 27.2 |
| IPM | 42.94 | 49.51 | 57.6 | 80.87 | 9.96 | 27.48 |
| GSS | 41.63 | 49.12 | 57.73 | 71.27 | 9.19 | 27.73 |
| Herding | 44.65 | 48.47 | 58.02 | 79.02 | 9.38 | 28.43 |
| LARS | 43.55 | 49.73 | 57.32 | 79.96 | 10.21 | 28.1 |
| SBS-fix | 44.87 | 50.39 | 58.89 | 81.16 | **12.78** | 29.37 |
| SBS | **45.64** | **53.71** | **61.13** | **82.22** | **12.78** | **29.41** |

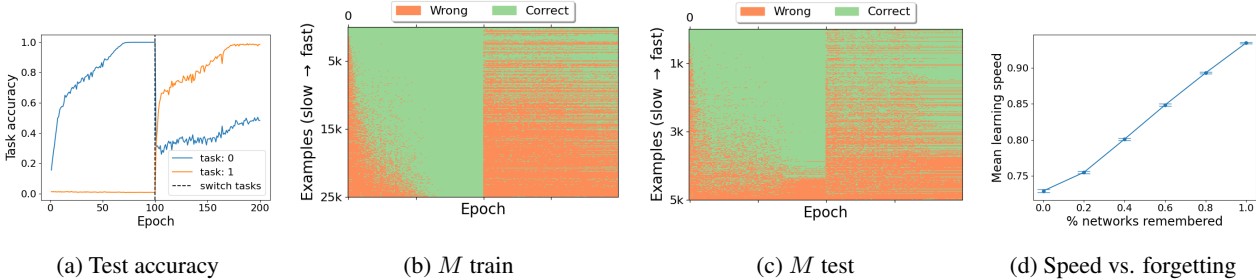

(a) Test accuracy      (b) $M$ train      (c) $M$ test      (d) Speed vs. forgetting

Figure 7: Repeating Fig. 2 in class incremental settings. All results are trained on CIFAR-100-2, without a replay buffer. (a) Mean test accuracy of each task. (b-c) first task's binary epoch-wise classification matrices $M$ for train and test data. (d) The mean learning speed of examples in the first task vs. the % of networks that remember them at the end of the second task.

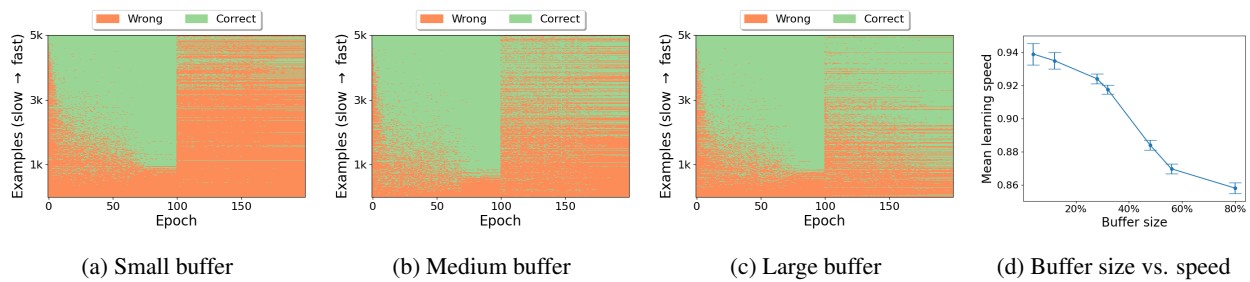

(a) Small buffer      (b) Medium buffer      (c) Large buffer      (d) Buffer size vs. speed

Figure 8: Repeating Fig. 3 for class incremental learning. Impact of the buffer size on forgetting dynamics. (a-c) The first task's binary matrices $M$, when adding replay buffers of varying buffer sizes. (d) The mean learning speed of remembered examples of models with different buffer sizes. Models with bigger replay buffers remember slower-to-learn examples.

## C. Correlations Between Learning Speed and Catastrophic Forgetting in Different Datasets

In Section 2, we established a link between the *learning speed* of examples and catastrophic forgetting, primarily utilizing ResNet-18 networks trained on CIFAR-100-2. Here, we expand upon these findings by demonstrating similar phenomena across different datasets, architectures, and task numbers.

### C.1. Other Datasets

We first explore other datasets using the experimental setup detailed in Fig. 2. We extend our analysis to CIFAR-10-2, CIFAR-100-2, and a subset of TinyImageNet-2 (comprising the initial 40 classes). The extended epoch-wise classification matrices for each task and the correlation between *learning speed* and the percentage of networks remembering each example are plotted in Fig. 11. Consistently across all three datasets, we observe a robust correlation between *learning speed* and the likelihood of example retention during continual training. Notably, faster-learned examples exhibit lower rates of catastrophic forgetting, as evidenced by both quantitative correlations and visual inspection of the extended epoch-wise classification matrices.

Further, in Fig. 12, we expand upon Fig. 3 to include CIFAR-10-2. This extension reveals that as the buffer size increases, continual models tend to retain slower-to-learn examples. Analogous to the results observed in CIFAR-100-2, as depicted in Fig. 3, we also observe a near-perfect negative correlation between the buffer size and the mean learning speed of remembered examples in CIFAR-10-2.

### C.2. Classification Matrices of Different Datasets

In Section 2, we analyze epoch-wise classification matrices, which indicate whether each example was classified correctly or incorrectly at each epoch during training. Figs. 2 and 3 plot these matrices for CIFAR-100-2 and CIFAR-10-2, revealing a clear correlation between learning speed and catastrophic forgetting: examples learned quickly are less likely to be forgotten during task switches.

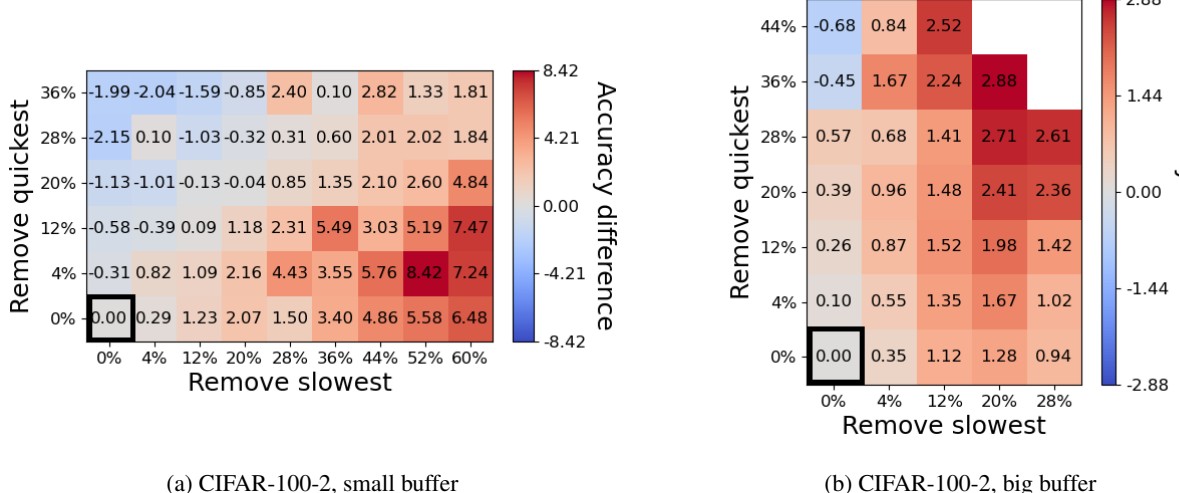

(a) CIFAR-100-2, small buffer

(b) CIFAR-100-2, big buffer

Figure 9: Repeating Fig. 4 for class incremental learning. Comparison of buffer compositions for various buffer sizes. (a) CIFAR-100-2 with a small $1k$ buffer. (b) CIFAR-100-2 with a large $10k$ buffer. Note that since class incremental learning is harder than task incremental learning, it is beneficial to remove fewer "quick" examples in this case.

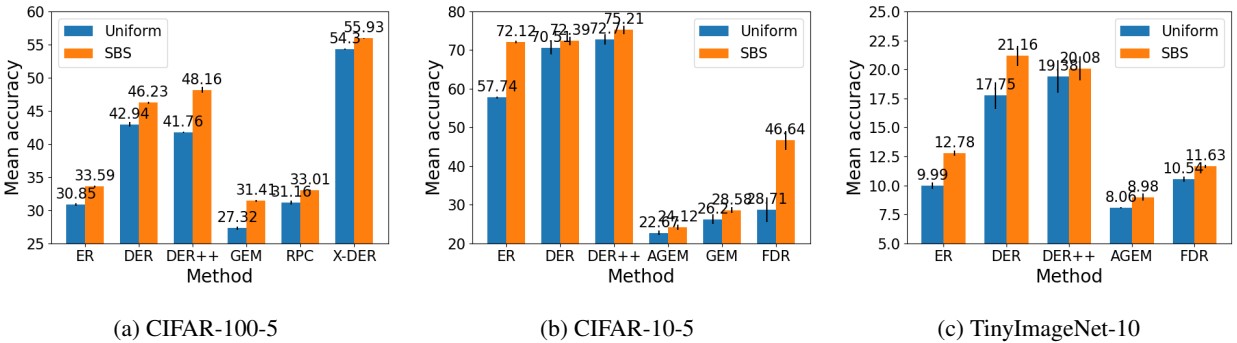

(a) CIFAR-100-5

(b) CIFAR-10-5

(c) TinyImageNet-10

Figure 10: SBS vs. uniform sampling with various continual learning methods, in class-incremental settings. Each bar shows the mean final test accuracy across all tasks for each method, with a uniform sampled buffer (traditionally done) in blue, and SBS in orange. Error bars denote the standard error.

In Fig. 15, we extend this analysis to the classification matrices of the first task of TinyImageNet-2, CIFAR-100-20, and CIFAR-10-5. For these datasets, which involve more than two tasks, the x-axis includes multiple task switches, marked by vertical dashed black lines. The same correlation is observed and becomes even more pronounced: examples from the first task are increasingly forgotten as additional tasks are introduced. With each new task, the newly forgotten examples are increasingly those learned earlier in training. Notably, examples learned fastest remain robust to forgetting, even after a long sequence of tasks, while those learned more slowly are more vulnerable.

### C.3. Other Architectures

We extend the results from Fig. 2 to architectures beyond ResNet-18 by creating two new, smaller variants: Small-ResNet and Tiny-ResNet, which can be found in Fig. 13. These are formed by reducing both the width and depth of ResNet-18 by a factor of 2 and 4 respectively. We replicate the experimental setup from Fig. 2, training 10 models of each architecture on CIFAR-100-2 without a replay buffer, and storing the extended epoch-wise classification matrix. The order examples appear in the matrix is sorted by the *learning speed* of the examples. All matrices depict the classification of the test dataset. In all architectures, *learning speed* is highly correlated with catastrophic forgetting: networks forget more examples learned later in training while retaining an almost perfect recollection of those learned early on. Additionally, stronger architectures,

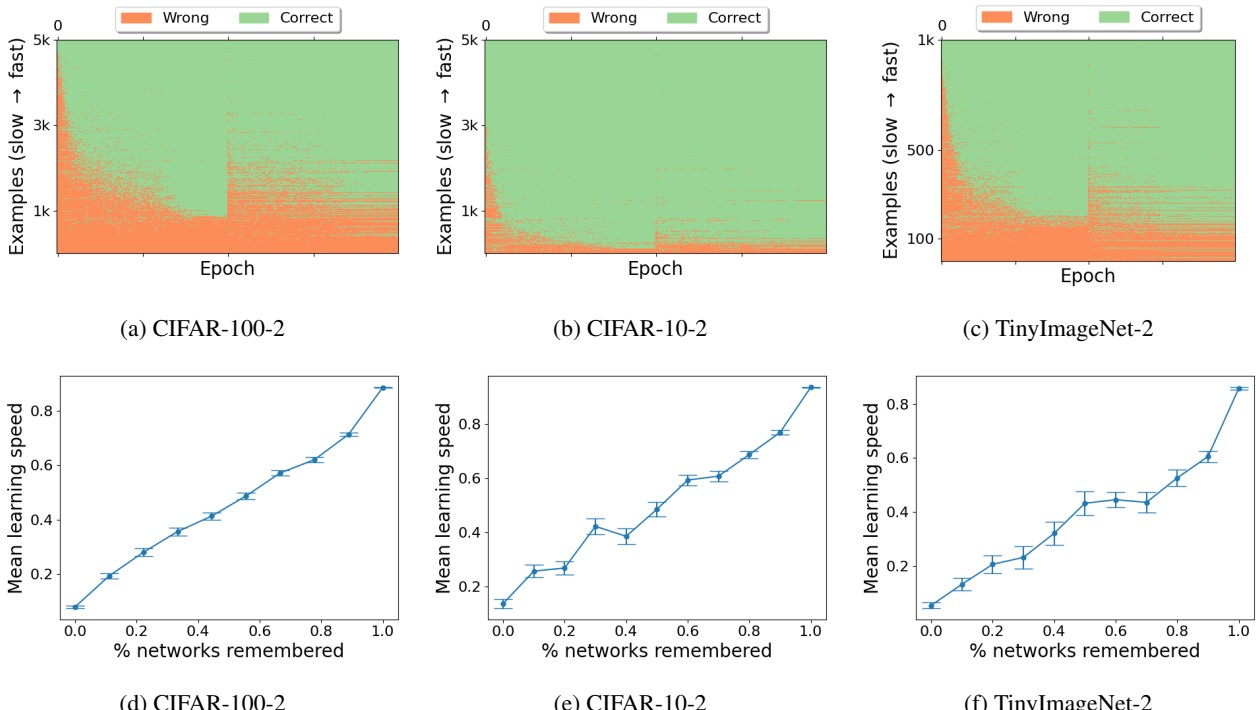

(a) CIFAR-100-2        (b) CIFAR-10-2        (c) TinyImageNet-2

(d) CIFAR-100-2        (e) CIFAR-10-2        (f) TinyImageNet-2

Figure 11: Extendeing Fig. 2 to CIFAR-100-2, Cifar-10-2 and a subset of TinyImageNet-2. (top) The first task's binary epoch-wise classification matrices $M$ for the test data of each dataset. The y-axis denotes examples, and the x-axis denotes epochs, indicating if an example is correctly classified by the model at the given epoch. The order in which examples appear is sorted by *learning speed*. Faster-learned examples from the first task are less likely to be forgotten at the end of the second task. (bottom) The mean *learning speed* of examples in the first task vs. the percentage of networks that remember them at the end of the second task. Networks forget more examples learned slowly across all 3 datasets.

similar to larger buffers, can remember slower-to-learn examples. This reinforces the motivation for SBS, as it shows that the examples selected by SBS are those the model could learn independently if it had a stronger architecture.

### C.4. Other Continual Learning Algorithms

In §3.2, we demonstrated the benefits of non-uniform buffer compositions for various continual learning methods. We extend these findings to additional algorithms including DER, DER++, GEM, A-GEM, RPC, and X-DER. For each algorithm, we trained 10 networks with different buffer compositions by varying the *quick* and *slow* hyperparameters of SBS. For each algorithm, we trained 10 networks on different buffer compositions, achieved by varying the *quick* and *slow* hyperparameters of SBS. In all cases, we used the hyperparameters and architectures suggested in the original works. We used a buffer of size 500, due to its popularity in previous works. Consistent with the GEM results, we found that replay buffers focused on examples learned mid-way through the training process were most beneficial. Additionally, in all cases, removing slower-to-learn examples proved advantageous, given the small buffer size. These results can be found in Fig. 14.

### C.5. Two Tasks Vs. Multi-Task

In the behavioral analysis, we focused on a two-task setting as it provides a simple and controlled environment. However, SBS was introduced in multi-task scenarios, which are standard in the CL literature and allow for more meaningful comparisons. For completeness, we now extend our SBS results to the two-task setting and the behavioral results to multi-task scenarios, aligning them with each other.

Fig.4 expands on Fig.6 by showcasing a broader range of SBS hyperparameters, hence including the results of SBS on these datasets. Using the RotNet auxiliary task to pick the hyperparameter in those cases obtains the hyperparameters that maximize performance across all three cases.

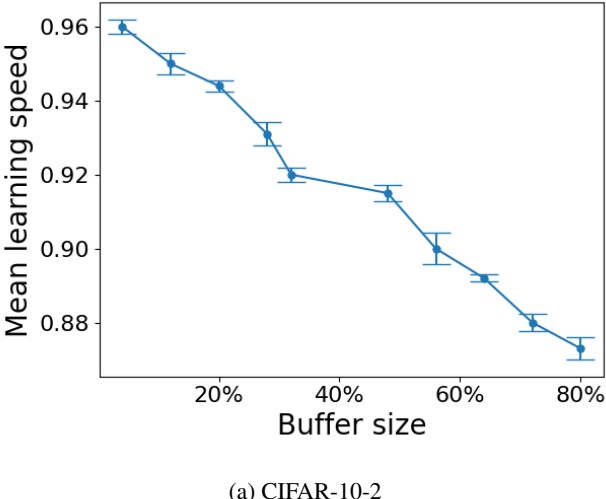

(a) CIFAR-10-2

Figure 12: Extending Fig. 3d to CIFAR-10-2. We plot the mean *learning speed* of remembered examples by 10 models trained with different buffer sizes. Models with bigger replay buffers remember slower-to-learn examples.

Table 3 extends Table 1 with results on CIFAR-10-2 and CIFAR-100-2, showing trends consistent with the multi-task setting. Additionally, Figs.16 and 17 extend Fig.4 to CIFAR-100-20 and CIFAR-10-5 (buffer size 5k) in both class- and task-incremental settings, showing similar qualitative trends.

Notably, while SBS improves performance in both two- and multi-task scenarios, its impact is even more pronounced in multi-task settings, further highlighting its practical value.

## D. The Effects of Different Learning Hyper-Parameters

In this section, we investigate how varying learning hyperparameters influences the optimal composition of examples in the replay buffer. We replicate the experiment shown in Fig. 4a, training networks on CIFAR-100-2 with a buffer size of 1000, while modifying factors such as network architecture, optimizer choice, learning rate, regularization strength, and the number of training epochs.

Across all configurations, we observe consistent qualitative trends similar to those in Fig. 4a: a wide range of buffer compositions significantly improves performance and alleviates catastrophic forgetting. These findings suggest that the conclusions drawn in the main paper are robust and generalizable, extending to continual learning scenarios under different hyperparameter settings.

**Optimizers.** In Fig. 18, we compare the performance of different buffer compositions when training ResNet-18 on CIFAR-100-2 using three optimizers: SGD, Adam, and Adagrad. For SGD, we used a momentum of 0.9, a weight decay of 0.0005, and a learning rate of 0.1, as is common for CIFAR-100 training. For Adagrad, we employed a learning rate of 0.01 and a weight decay of 0.0001. For Adam, we used a learning rate of 0.001, a weight decay of 0.0001, and (0.9, 0.999) betas. These hyperparameters were tuned to optimize ResNet-18's performance on CIFAR-100 without considering continual learning, as the optimal settings vary by optimizer due to their differing update mechanisms.

Our results indicate that similar buffer compositions consistently enhance performance across all three optimizers. This consistency suggests that the benefits of our analysis are robust to the choice of optimizer, further supporting the generalizability of our findings.

**Architectures.** In Fig. 19a, we evaluate the impact of buffer composition on the performance of three different architectures: ResNet-18, VGG-16, and a smaller version of ResNet, tiny-ResNet. All models were trained on CIFAR-100-2 under identical conditions. Our results show that similar buffer compositions consistently improve learning performance across these diverse architectures. This finding suggests that our analysis is not specific to any single architecture.

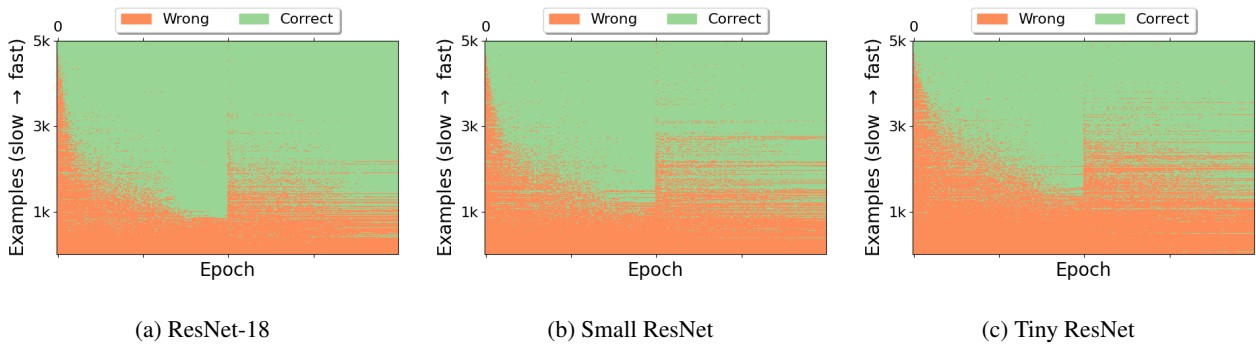

Figure 13: Comparing the extended epoch-wise classification matrix for different architectures, the y-axis represents examples, and the x-axis represents epochs, indicating if an example is correctly classified by the model in a given epoch. In (a), we train ResNet-18. In (b), we train ResNet-18 with both width and depth reduced by half. In (c), we train ResNet-18 with both width and depth reduced by a factor of four. In all cases, the order examples appear is sorted by *learning speed*. Examples learned quickly in the first task are less likely to be forgotten after the second task.

**Training epochs.** In Fig. 20, we compare the performance of models trained for different numbers of epochs per task, all using a cosine learning rate scheduler to ensure exposure to both large and small learning rates throughout training. While reducing the number of epochs leads to a general drop in performance, we observe that similar buffer compositions consistently yield strong results, even under shorter training schedules. It is worth noting that the accuracy of determining the *learning speed* of each example may decrease with fewer epochs, as there is less time for the model to adapt. Nonetheless, the qualitative trends remain robust, with specific buffer compositions consistently outperforming or underperforming the random baseline across all tested epoch counts.

**Learning Rates.** We examine the impact of learning rates on model performance using SGD with rates of 0.05 and 0.2 (Figs. 21a, 21b). While the final accuracies differ across these configurations, the qualitative behavior remains consistent: the same buffer composition consistently achieves strong results.

**Fine-Tuning.** In multi-task training, it is common to reduce the learning rate for subsequent tasks to fine-tune the network on the next task. To investigate this, we replicate the experiment in Fig. 4a, lowering the learning rate for the second task by a factor of 10. The results align closely with those in Fig. 4a, indicating that our findings are robust and not solely attributable to the specific learning rates used in the original experiments. These results can be found in Fig. 21c.

**Regularization.** All ResNet-18 experiments in the main paper were conducted with a small weight decay of 0.0005. To assess the impact of this regularization, we replicate the experiment from Fig. 4a in Fig. 19b where we remove the weight decay entirely. The results remain qualitatively consistent, indicating that the conclusions from our analysis are not sensitive to this specific regularization choice.

**Different splits of CIFAR.** We repeated the experiment from Fig. 4a using a different split of CIFAR-100, where classes were randomly assigned to tasks. The results, shown in Fig. 19c, remain consistent with those in Fig. 4a. Importantly, all task splits in our experiments were across the paper chosen arbitrarily, ensuring unbiased evaluations.

## E. Standard Errors

In the main paper, we omitted the standard errors for visualization porpuses from Figs. 4,5 and Table 1. In all cases, these were usually very small and did not affect the qualitative results. For completeness, we report the omitted values in this section. The standard error for Figs. 4,5 can be found in Figs. 22,23 respectively. The standard errors of Table 1 can be found in Table 4.

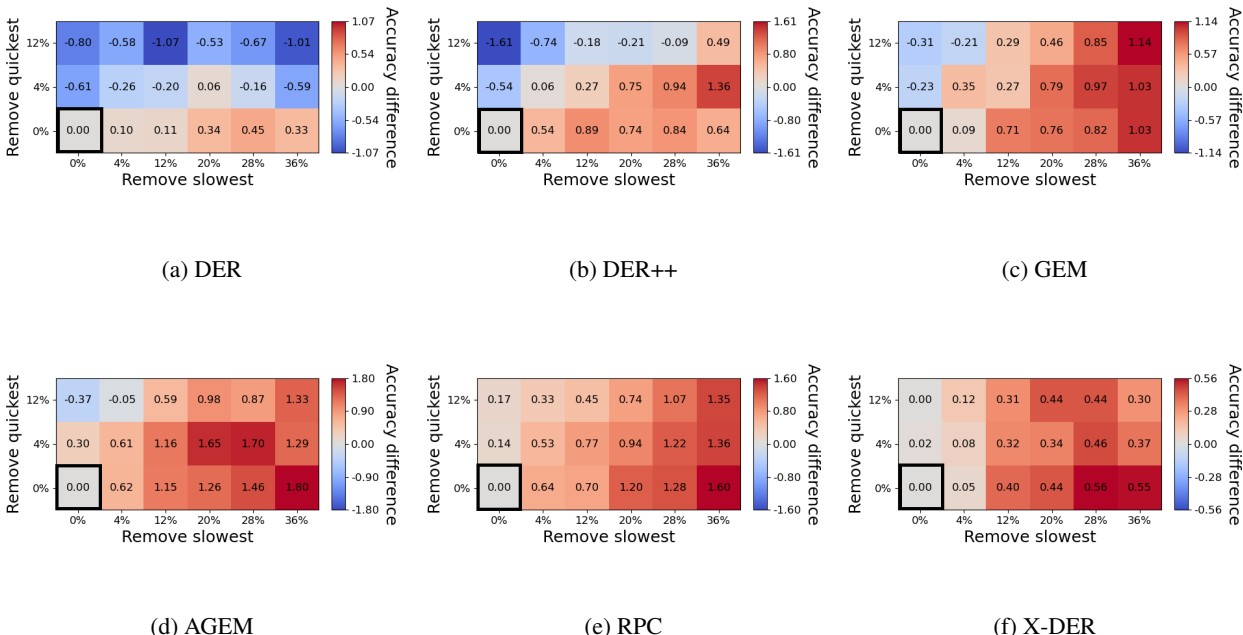

Figure 14: Comparing different replay buffer compositions for various continual learning methods, with a buffer of size 500. The experimental setup replicates the one done in Fig. 4. Like the experience replay case, focusing on examples learned midway through the learning process is most beneficial. Due to the smaller buffer, removing slower examples is better.

## F. Independency of the Optimal Buffer Composition When Changing the Subsequent Tasks

We empirically examine how subsequent tasks impact the ideal composition of a replay buffer for a given task. We find that regardless of the similarity or dissimilarity between subsequent tasks and the original task, the optimal replay buffer composition remains largely independent and consistent. To show this, we conduct experiments with three distinct tasks denoted A, B, and C. By training the model on task A and subsequently introducing either task B or task C while keeping the original task unchanged, we analyze the ideal replay buffer composition under varying subsequent tasks.

In Section 2.6, we demonstrate that when selecting tasks A, B, and C from the same dataset, the optimal buffer composition of task A remains unaffected by tasks B or C. We observe consistent quantitative behavior across different *quick* and *slow* parameters. We further explore task variations by setting A as the first 50 classes of CIFAR-100, B as the last 50 classes of CIFAR-100, and C as a rotation classification (Gidaris et al., 2018) task. In the rotation classification task, examples from A are randomly rotated by angles of $\{90°, 180°, 270°\}$, with labels adjusted accordingly. We evaluate the performance of 10 networks trained continuously with different buffer compositions first on task A and then on task B (Fig. 24a), and on task A followed by task C (rotation) (Fig. 24b). Consistent with previous findings, the performance of various buffer compositions remains consistent, independent of the choice of B or C. This property enables us to utilize the rotation task for hyper-parameter grid search without requiring additional label data.

We further modify task C to comprise examples from the last 50 classes of CIFAR-100 but with random labels (Zhang et al., 2021). Training on random labels forces the network to memorize the examples, as no generalization is possible, eliminating any potential overlap between the subsequent tasks. The results of training on task A followed by the random label task are depicted in Fig. 24c. Analogous to the rotation classification scenario, we observe that the same buffer compositions of task A remain effective, further suggesting that the optimal buffer composition of A does not depend on the subsequent task.

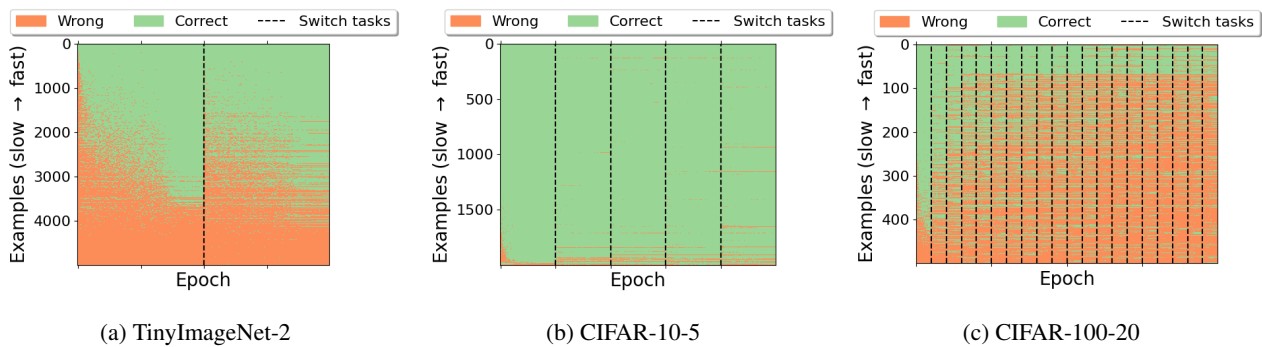

(a) TinyImageNet-2  (b) CIFAR-10-5  (c) CIFAR-100-20

Figure 15: Repeating Fig. 2c with different datasets and task counts: (a) TinyImageNet with 2 tasks, (b) CIFAR-10 with 5 tasks, and (c) CIFAR-100 with 20 tasks. Dashed black lines indicate the introduction of a new task. Consistent with Fig. 2c, examples learned more slowly are more prone to forgetting after task transitions. As the number of tasks increases, even examples learned relatively quickly become susceptible to forgetting.

Table 3: Extended results from Table 1 for CIFAR-10-2 and CIFAR-100-2. For convenience, the original results from Table 1 are included. Standard errors are reported separately in Table 4.

|  | CIFAR-10-2 | | CIFAR-100-2 | | CIFAR-100-20 | | CIFAR-10-5 | | TinyImageNet-2 | |
| Buffer size | $1k$ | $10k$ | $1k$ | $10k$ | $1k$ | $10k$ | $1k$ | $10k$ | $1k$ | $10k$ |
|---|---|---|---|---|---|---|---|---|---|---|
| Random | 87.67 | 95.03 | 69.03 | 79.24 | 51.75 | 71.25 | 79.48 | 82.4 | 49.81 | 61.48 |
| Max entropy | 84.44 | 94.63 | 64.27 | 77.91 | 48.3 | 69.83 | 77.28 | 81.72 | 44.49 | 57.88 |
| IPM | 85.9 | 94.76 | 64.16 | 75.73 | 49.91 | 71.7 | 79.3 | 80.57 | 48.58 | 61.97 |
| GSS | 85.28 | 95.55 | 64.15 | 80.11 | 48.13 | 72.9 | 76.71 | 84.13 | 49.36 | 62.89 |
| Herding | 88.41 | 94.53 | 71.07 | 78.97 | 54.0 | 73.51 | 81.8 | 81.03 | 51.17 | 62.44 |
| LARS | 87.43 | 95.43 | 69.53 | 79.92 | 51.82 | 71.41 | 79.17 | 84.93 | 50.22 | 62.78 |
| SBS-fix | 88.11 | 95.56 | 70.29 | 80.2 | 54.65 | 73.72 | 82.24 | 86.15 | **52.15** | 63.12 |
| SBS | **88.99** | **96.03** | **71.71** | **80.62** | **55.43** | **74.80** | **83.59** | **89.17** | **52.15** | **63.16** |

## G. Relationship Between Other Scoring Methods and the Learning Speed

To analyze catastrophic forgetting, we focus on its relationship with learning speed (Eq.1). While other metrics exist—such as uncertainty score (Bang et al., 2021), which evaluates classification consistency under augmentations, and c-score (Jiang et al., 2020), which measures expected accuracy on held-out instances – they incur higher computational costs. Using per-example loss as a score is computationally simpler but shows a weak correlation with catastrophic forgetting. As shown in Fig.25, examples sorted by uncertainty and c-score show a moderate correlation with forgetting, while learning speed provides the strongest correlation. Therefore, we adopt learning speed throughout this work for its computational efficiency and strong alignment with catastrophic forgetting.

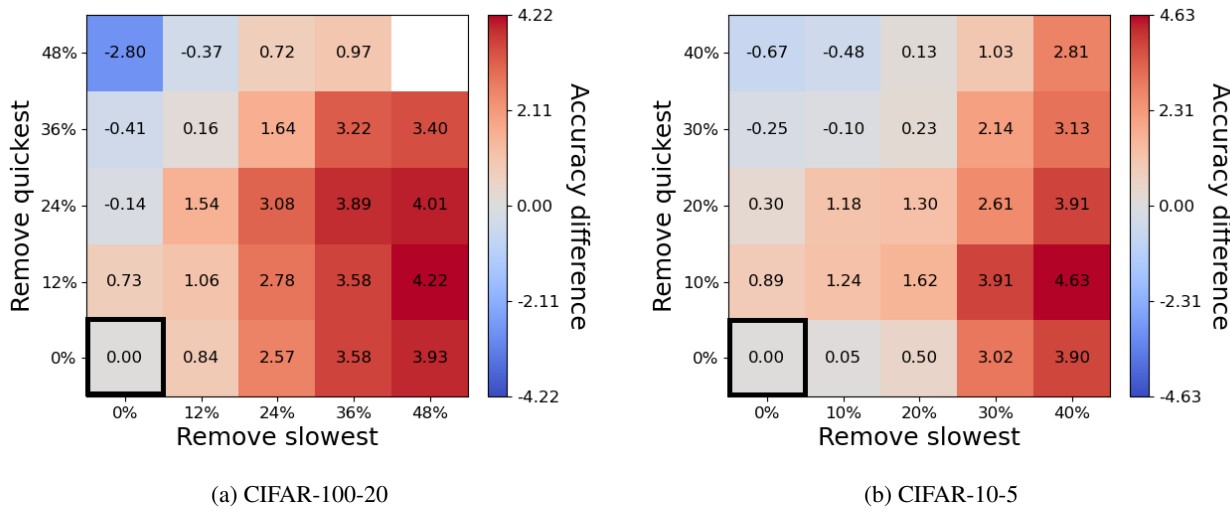

(a) CIFAR-100-20

(b) CIFAR-10-5

Figure 16: Comparison of buffer compositions in multi-task training scenarios: (a) CIFAR-100-20, (b) CIFAR-10-5, both trained with a 5k buffer. Similarly to the 2-task case, a wide range of buffer compositions significantly improves performance, helping to mitigate catastrophic forgetting. The performance gains of Goldilocks tend to get bigger in the multi-task case.

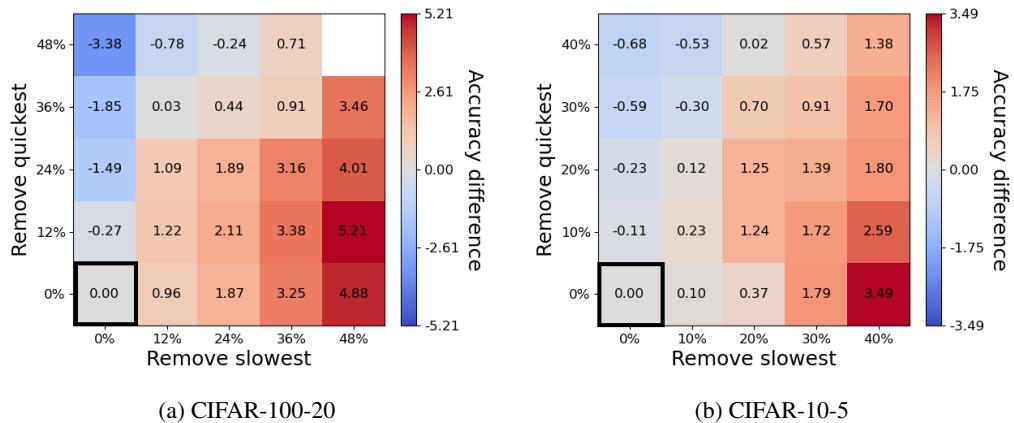

(a) CIFAR-100-20

(b) CIFAR-10-5

Figure 17: Repeating Fig. 16 for class incremental learning. (a) CIFAR-100-20, (b) CIFAR-10-5, both with a 5k buffer. Similar to the 2-task case, a wide range of buffer compositions significantly improves performance, helping to mitigate catastrophic forgetting.

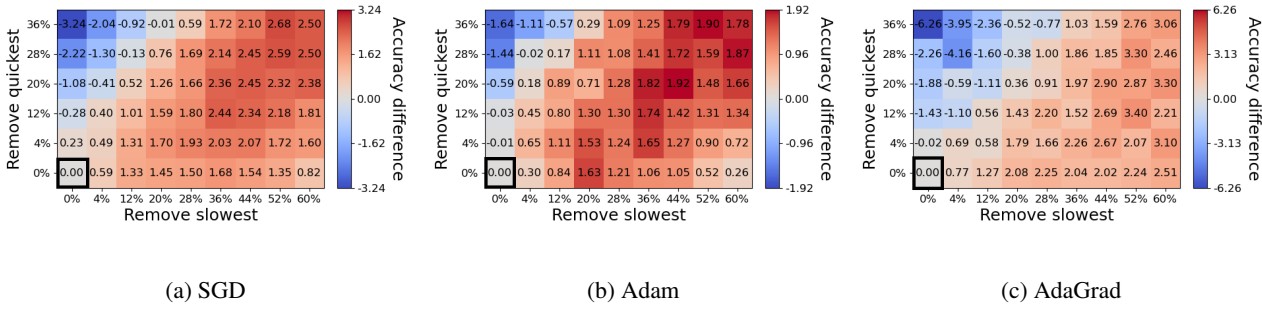

(a) SGD

(b) Adam

(c) AdaGrad

Figure 18: Repeating Fig. 4a using different optimizers: (a) SGD, (b) Adam, and (c) AdaGrad. Across all optimizers, a wide range of buffer compositions consistently enhances performance, yielding similar qualitative results to the experiment in the main text.

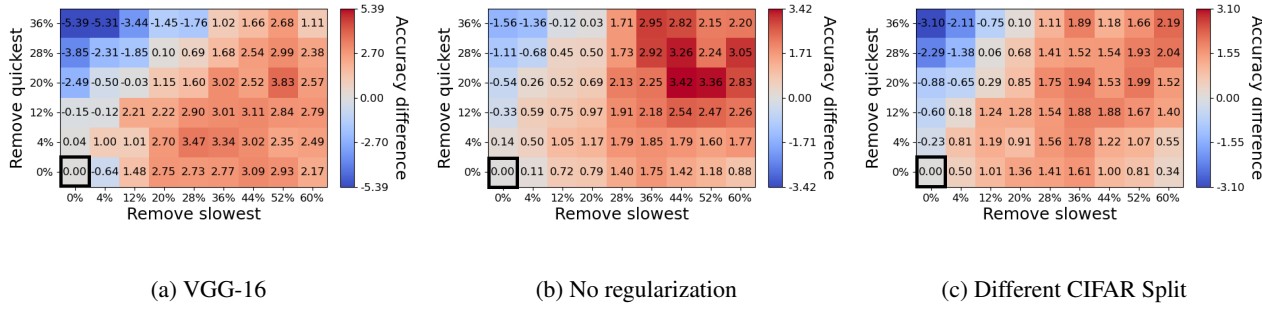

(a) VGG-16                (b) No regularization           (c) Different CIFAR Split

Figure 19: Repeating Fig. 4a under different learning settings: (a) replacing the ResNet-18 architecture with VGG-16, (b) removing weight decay from ResNet-18, and (c) using a random class-to-task split of CIFAR-100-2. In all cases, the results remain qualitatively similar, highlighting the robustness of the analysis.

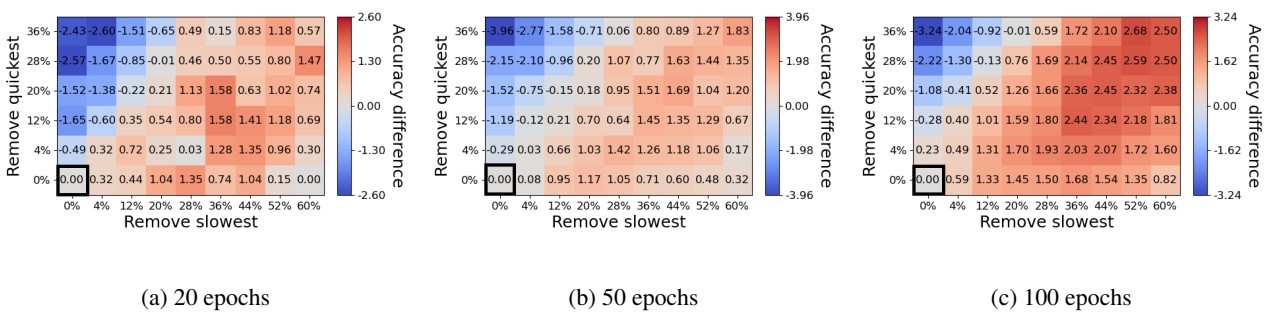

(a) 20 epochs                 (b) 50 epochs                (c) 100 epochs

Figure 20: Repeating Fig. 4a with varying training epochs: (a) 20 epochs, (b) 50 epochs, and (c) 100 epochs per task. Despite shorter training durations, the qualitative results remain consistent, with a wide range of buffer compositions significantly improving performance.

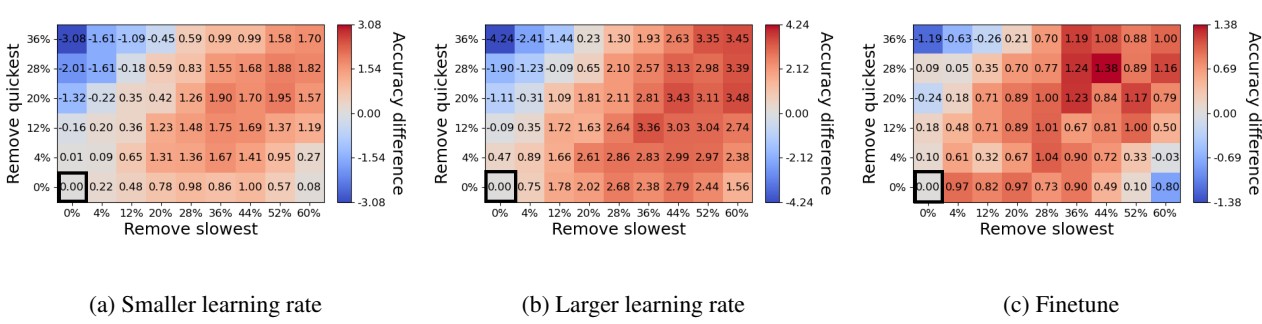

(a) Smaller learning rate        (b) Larger learning rate          (c) Finetune

Figure 21: Repeating Fig. 4a with varying learning rates: (a) doubling the learning rate, (b) halving the learning rate, and (c) keeping the first task's learning rate unchanged while reducing the second task's by a factor of 10. These scenarios, common in continual learning research, yield consistent qualitative results, indicating that our analysis is not dependent on a specific learning rate.

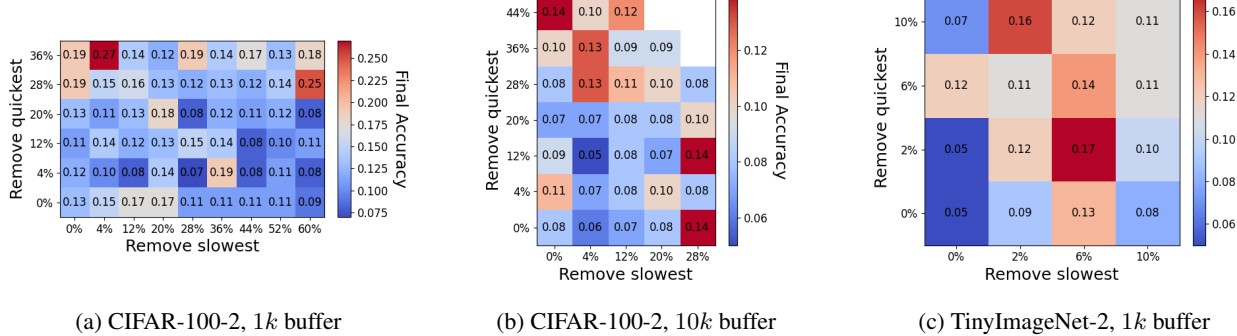

(a) CIFAR-100-2, $1k$ buffer      (b) CIFAR-100-2, $10k$ buffer      (c) TinyImageNet-2, $1k$ buffer

Figure 22: Standard errors for Fig. 4. The error is taken over 10 repetitions in each experiment.

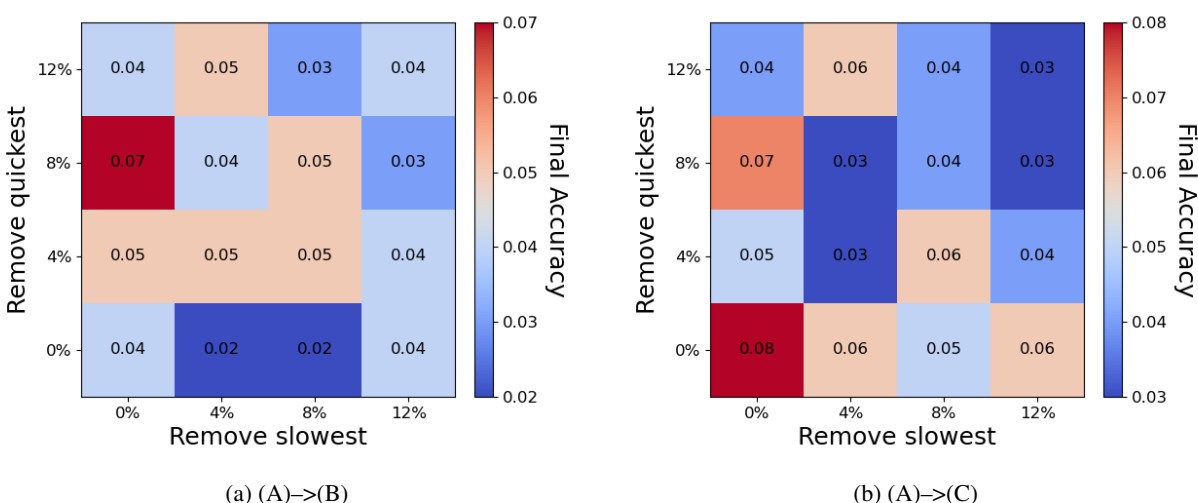

(a) (A)–>(B)      (b) (A)–>(C)

Figure 23: Standard errors for Fig. 5. The error is taken over 10 repetitions in each experiment.

Table 4: Standard error of Table 1.

|  | CIFAR-10-2 | | CIFAR-100-2 | | CIFAR-100-20 | | CIFAR-10-5 | | TinyImageNet-2 | |
|---|---|---|---|---|---|---|---|---|---|---|
| Buffer size | $1k$ | $10k$ | $1k$ | $10k$ | $1k$ | $10k$ | $1k$ | $10k$ | $1k$ | $10k$ |
| Random | 0.06 | 0.05 | 0.22 | 0.12 | 0.18 | 0.04 | 0.1 | 0.11 | 0.15 | 0.07 |
| Max entropy | 0.23 | 0.06 | 0.23 | 0.15 | 0.12 | 0.02 | 0.1 | 0.03 | 0.2 | 0.13 |
| IPM | 0.16 | 0.13 | 0.29 | 0.16 | 0.17 | 0.06 | 0.15 | 0.06 | 0.15 | 0.13 |
| GSS | 0.13 | 0.08 | 0.29 | 0.16 | 0.15 | 0.01 | 0.11 | 0.04 | 0.13 | 0.07 |
| Herding | 0.09 | 0.12 | 0.24 | 0.14 | 0.11 | 0.1 | 0.14 | 0.06 | 0.16 | 0.11 |
| LARS | 0.15 | 0.12 | 0.27 | 0.18 | 0.16 | 0.04 | 0.11 | 0.05 | 0.13 | 0.08 |
| SBS-fix | 0.08 | 0.1 | 0.25 | 0.11 | 0.08 | 0.07 | 0.13 | 0.11 | 0.19 | 0.1 |
| SBS | 0.03 | 0.1 | 0.22 | 0.1 | 0.1 | 0.02 | 0.1 | 0.12 | 0.16 | 0.1 |

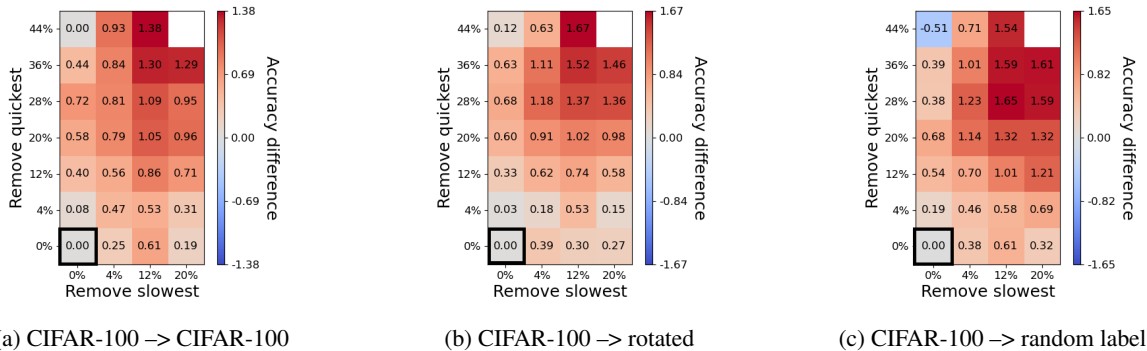

(a) CIFAR-100 –> CIFAR-100        (b) CIFAR-100 –> rotated        (c) CIFAR-100 –> random labels

Figure 24: Comparing replay buffer compositions when training the same original task, followed by different subsequent tasks. Each entry in the matrices denotes the difference in final accuracy between each buffer composition and a buffer sampled uniformly at random. Each composition is repeated 10 times. Networks are trained using experience replay on 2 tasks, with a buffer size of $10k$. This Figure replicated the same experimental setup as Fig. 5. In (a) we train the models on the 50 first classes of CIFAR-100, followed by the last 50 classes of CIFAR-100. In (b) we train the models on the 50 first classes of CIFAR-100, followed by a rotation classification task on the same examples (see text). In (c) we train the models on the 50 first classes of CIFAR-100, followed by the last 50 classes of CIFAR-100, but with random labels. In all cases, the same buffer compositions of the first task give the same qualitative results, suggesting that the optimal composition of the buffer of the first task is independent of the subsequent task.

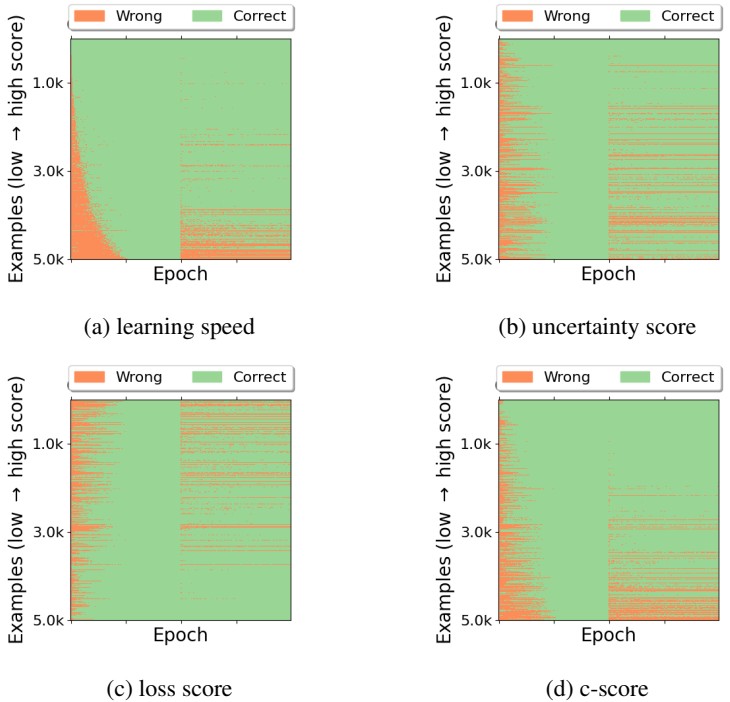

(a) learning speed        (b) uncertainty score

(c) loss score        (d) c-score

Figure 25: Comparison of the extended epoch-wise classification matrix for CIFAR-100-2, sorted by different example-level scoring functions (see App. G). (a) Learning speed (used throughout the paper), (b) Uncertainty measure from Bang et al. (2021), (c) Final loss, (d) c-score from Jiang et al. (2020). While all scores show some correlation with forgetting, as more complex examples tend to be forgotten more, learning speed shows the strongest, suggesting it is most suitable for our analysis.

