# OpenReview forum: "Predicting the Susceptibility of Examples to Catastrophic Forgetting"
_ICML.cc/2025/Conference — ICML 2025 poster_

### Official Review · Reviewer_f954 · 2025-03-01

**Overall Recommendation:** 3

**Summary:**

This paper reports on a large volume of observations regarding the learning speed and catastrophic forgetting in the context of continual learning, and proposes a new sampling strategy called SBS to improve the replay-based continual learning methods. The experiments consistently show the effectiveness of the proposed speed-based sampling, which highlights the practical value of the proposed strategy.

**Claims And Evidence:**

Most of the claims are based on empirical study and I don't have any objections on them. However, as they are based on the empirical observation, the academic value of the work is clumsy. Especially, the sizes of q and s should be more theoretically derived for the method.

**Essential References Not Discussed:**

Most of the important references are discussed in the paper.

**Experimental Designs Or Analyses:**

The overall design of the experiments and analyses follows the standard convention.

**Methods And Evaluation Criteria:**

The datasets used are popular in the field of continual learning, and all the settings follow the standard convention. However, in some experiments, the hyperparameters need to be justified objectively.

**Other Comments Or Suggestions:**

None

**Other Strengths And Weaknesses:**

The strongest point of the paper is an extensive empirical study and interesting observation. However, despite the extensive efforts, the academic value is limited to the empirical study. The authors should investigate more in depth on the proposed method, and need to provide a theoretical justification on the proposed strategy.

**Questions For Authors:**

- How can you determine the optimal values for q and s without such a arbitrary task?

- What is the reason of the learning speed for the catastrophic forgetting?

**Relation To Broader Scientific Literature:**

This is specific to the continual learning and less room for broader scientific literature.

**Theoretical Claims:**

There's no theoretical claim in this paper.

---

> ### Author Rebuttal · Authors · 2025-03-31
>
> Thank you for your thoughtful review. We appreciate your recognition of our extensive empirical study and the practical value of our findings.
>
> Regarding the connection between learning speed and catastrophic forgetting, our primary contribution is identifying and characterizing this phenomenon. While a formal theoretical analysis is an exciting future direction, many fundamental insights in machine learning (e.g., double descent, simplicity bias) were first observed empirically before theory followed. Despite the lack of theory, our results have several more intuitive explanations: later-learned examples likely depend on more complex or composite features, making them more susceptible to forgetting when new tasks are introduced, as those features are more likely to break first. Another intuition is that a similar pattern occurs in human learning, where foundational skills persist longer than more complex ones.
>
> On the choice of hyperparameters $q$ and $s$, our goal with SBS is to demonstrate that leveraging learning speed can improve continual learning across different algorithms and settings. While a more theoretically derived selection method could be valuable, it is non-trivial and beyond this paper’s scope. Instead, we propose a practical heuristic -- using a related self-supervised task (e.g., RotNet) to tune these parameters without additional labels. While RotNet was chosen arbitrarily, our experiments show that this approach consistently finds the best hyperparameters across all the setups we evaluated.

---

> > ### Comment · Reviewer_f954 · 2025-04-02
> >
> > I agree that theoretical justification may follow the empirical study, and in terms of completeness I wanted to note the necessity of the work for the future. In that sense, this work is on its way to that direction, and a little premature to the goal.

---

### Official Review · Reviewer_hXyj · 2025-03-06

**Overall Recommendation:** 2

**Summary:**

The manuscript addresses the challenge of selecting the most relevant examples to store in a memory buffer for rehearsal-based continual learning. Based on a preliminary analysis on the speed at which examples are learned and forgotten, the manuscript finds that the most complex samples are the fastest to be forgotten. Based on this, it introduces Speed-Based Sampling (SBS), a sampling strategy that excludes the $q$ fastest and $s$ slowest learned samples from the selection.

**Claims And Evidence:**

The claims are supported by a thorough analysis.

**Essential References Not Discussed:**

N/A

**Experimental Designs Or Analyses:**

While the design of the benchmarks is adequate, I believe the manuscript should compare against other methods that seek to retain complex or diverse samples from the stream, such as [1,2]. Indeed, while the proposal addresses this by measuring the "learning speed" of each sample, [1] uses the loss of the examples as a proxy for complexity.

[1]: Buzzega, Pietro, et al. "Rethinking experience replay: a bag of tricks for continual learning." ICPR 2021.
[2]: Bang, Jihwan, et al. "Rainbow memory: Continual learning with a memory of diverse samples."  CVPR 2021.

**Methods And Evaluation Criteria:**

The benchmarks are adequate.

**Other Comments Or Suggestions:**

N/A

**Other Strengths And Weaknesses:**

1: As mentioned in lines 389-391, the results presented in Tab. 1 for the proposed SBS are obtained with the RotNet auxiliary task. However, given the high computational cost of training the network multiple times, a more fair comparison should compare with the fixed values for q and s, as mentioned in l379. Indeed, while the manuscript mentions that "q=s=20% consistently enhances performance across all evaluated datasets", I did not find results to support such a claim. On the same note, I suggest including the computational cost (in terms of compute time) of SBS with the RotNet auxiliary task.

2: It was not clear to me if the experiments of Tab. 1 (main paper) and 2 (supplementary) were conducted on the TIL or CIL setting. From Sec 2.2 it seems to me that both figures and tables in the main paper are for TIL and that results for CIL are in the supplementary. However, App. B only mentions results for the qualitatives and the figures and I could not find the results for the main experiments in the CIL setting.

3: I did not understand the reasoning behind removing the $s$ slowest-learned examples from the dataset. According to the preliminary analysis, if the motivation is to retain the most complex samples, why would we need to discard the ones that would be the first to be forgotten?

**Questions For Authors:**

Overall the manuscript is well written and mostly easy to read and understand. I would consider raising my score upon addressing the concerns in “Other Strengths And Weaknesses" and the lack of similar methods that seek to retain complex or diverse samples from the stream.

**Relation To Broader Scientific Literature:**

The considerations regarding the speed at which the examples are learned are based on existing literature and the results are known in literature [1,2]. However, the proposed SBS seems novel and relevant for the CL field.

[1]: Maini, Pratyush, et al. "Characterizing datapoints via second-split forgetting." aNeurIps 2022.
[2]: Millunzi, Monica, et al. "May the Forgetting Be with You: Alternate Replay for Learning with Noisy Labels." BMVC 2024.

**Theoretical Claims:**

N/A

---

> ### Author Rebuttal · Authors · 2025-03-31
>
> Thank you for your thoughtful review and for your willingness to consider raising your score. Below, we address your concerns in detail.
>
> **concern 1**
>
> The advantages of picking $q=s=20\%$ across different settings are scattered throughout the figures of the paper: see Figs 4(a-b), 16(a-b), 17(a-b), 18(a-c), 19(a-c), 20(a-c), 21(a-c) and 24(a-c). However, we will add a dedicated appendix section explicitly showcasing results for picking $q=s=20\%$, hopefully making this point clearer.
>
> Additionally, we will add a new row to Table 1, reporting the results of SBS with these fixed hyperparameters, showing the advantages that can be gained by SBS with and without the hyper-parameter search. The results that will be added are:
>
> |buffer|1k|10k|
> |-|:-:|:-:|
> |CIFAR-100-20|54.65|73.72|
> |CIFAR-10-5|82.24|86.15|
> |TinyImageNet-2|52.15|63.12|
>
> to compare, the baseline results of random sampling (Table 1 in the original manuscript) are:
> |buffer|1k|10k|
> |-|:-:|:-:|
> |CIFAR-100-20|51.75|71.25|
> |CIFAR-10-5|79.48|82.4|
> |TinyImageNet-2|49.81|61.48|
>
> While removing hyperparameter tuning slightly reduces performance, SBS still consistently outperforms other methods across all settings. This demonstrates that SBS is effective even without extensive tuning.
>
> We will also add computational cost for both the RotNet-based version and the fixed-hyperparameter version in the appendix to provide a complete picture.
>
> **concern 2**
>
> Thank you for catching this oversight. You are correct that Table 1 (main paper) and Table 2 (supplementary) both report results for TIL. We also conducted experiments for CIL, and while they were qualitatively similar (though lower in absolute performance due to the increased difficulty of CIL), they were mistakenly omitted from App. B.
>
> We will correct this in the camera-ready version by including the CIL results in App. B.
>
> **concern 3**
>
> This is an interesting question. While we do not have a formal theoretical proof, we can provide an intuitive explanation based on our observations.
>
> While the training error of networks tends to go to $0$, meaning that every example in the dataset is going to be learned at some speed, the test error often does not. The training examples that are learned slowest are often those that correspond to points near test points that the model simply did not learn, meaning they are inherently difficult for the model to generalize to, even in non-continual settings. Therefore, in the CL settings, where the replay buffers are limited, we want to focus on examples that the model can generalize from, and removing the slowest-learned examples helps allocate space to more useful ones, improving overall performance. However, we note that the better the model can perform on the original task, the smaller the number of these unhelpful slower-to-learn examples, allowing us to remove less of them to get better performance.
>
> **comparison to other methods**
>
> You suggested comparisons with methods that retain complex or diverse samples, specifically Rainbow Memory (RM) [1] and LARS [2]. Below, we summarize how our method relates to them and provide additional quantitative comparisons.
>
> **Rainbow Memory (RM)**
>
> The original RM paper focuses on blurry-CIL settings, whereas our study considers disjoint settings. The RM paper itself notes that RM does not consistently improve performance in disjoint settings, often performing similarly to random sampling. Since SBS outperforms random sampling, this suggests that SBS is superior to RM in these settings.
>
> That said, we acknowledge RM’s potential value and have already discussed its relation to SBS in App. G. We will make this connection more explicit in the main paper and include RM results in Table 1 for direct comparison.
>
> **LARS (from BAGS of Tricks)**
>
> LARS is presented in [2] as one of several "tricks" to improve CL and was not presented as a stand-alone sampling method, which is expected to improve performance across different settings. LARS suggests to sample the buffer randomly, but to remove examples from it in a suggests loss-aware, where examples with low loss will be removed more frequently. We agree that adding a comparison to LARS can strengthen our work, and we added such a comparison to the paper. To compare to SBS, we isolated LARS from the rest of the "tricks" suggested in [1], and evaluated it on the different datasets and buffer sizes in Table 1. Below are the results:
>
> |buffer|1k|10k|
> |-|:-:|:-:|
> |CIFAR-100-20|51.82|71.41|
> |CIFAR-10-5|79.17|84.93|
> |TinyImageNet-2|50.22|62.78|
>
> While LARS improves over random sampling in most cases, its gains are smaller than those achieved by SBS. Following your suggestion, we will add both RM and LARS results to Table 1 in the camera-ready version for a clearer comparison.
>
> ---
> [1] Bang, Jihwan, et al. Rainbow memory: Continual learning with a memory of diverse samples. (CVPR 2021)
>
> [2] He, Tong, et al. Bag of tricks for image classification with convolutional neural networks. (CVPR 2019)

---

### Official Review · Reviewer_98Km · 2025-03-11

**Overall Recommendation:** 3

**Summary:**

In this work, the authors investigate catastrophic forgetting from a behavioral perspective,  observing the connection between learning speed and forgetting: examples learned more quickly tend to be more resistant to forgetting. Motivated by the observation, this paper introduces Speed-Based Sampling (SBS), a simple yet general strategy that selects replay examples based on their learning speed, for replay-based continual learning. Experiments show the advantages of the proposed SBS over uniform sampling and other sampling baselines.

**Claims And Evidence:**

Yes.

**Essential References Not Discussed:**

NA

**Experimental Designs Or Analyses:**

Note that the paper highlights the benefits of SBS to existing replay-based continual learning methods. However, the authors only compare their method with uniform sampling when utilizing continual learning methods. To make the claim sounder, the authors should also compare it with other sampling methods when utilizing replay-based continual learning methods.

**Methods And Evaluation Criteria:**

Yes.

**Other Comments Or Suggestions:**

Refer to Weaknesses part.

**Other Strengths And Weaknesses:**

Strength:

1.	The phenomenon that examples learned later are more likely to be forgotten while earlier learned examples are not is well studied and supported by extensive experiments.

Weaknesses:

1.	The proposed method may be invalid for prevailing large-scale training. First, when training with large-scale data, people usually train on the dataset one or two epochs, making the proposed ‘learning speed score’ inaccurate as mentioned by the authors. Second, with large-scale data set, it would be computationally expensive to run the RotNet auxiliary task on the same dataset for selecting hyperparameters q and s to get the best performance shown in the paper. Although the authors suggest that simply setting q = s = 20% is generally good choice, the performance of the choice is not clear in the paper.

2.	The paper emphasizes the benefits of SBS to existing replay-based continual learning methods, but the authors only compare their method with uniform sampling when utilizing continual learning methods.

**Questions For Authors:**

1.	In Table 1, methods ‘Max Ent’ and ‘IPM’ are almost consistently worse than random sampling, which seems unusual. Do you have any insights into this phenomenon?

**Relation To Broader Scientific Literature:**

The proposed sampling method is tailored for continual learning tasks.

**Theoretical Claims:**

No, as no theoretical claims are included.

---

> ### Author Rebuttal · Authors · 2025-03-31
>
> Thank you for your thoughtful review. We appreciate that you found our analysis comprehensive and our results well-supported. Below, we address your comments in detail:
>
> **Comparison with non-uniform sampling in continual learning methods**
>
> Most competitive continual learning methods rely on random sampling [1] because non-uniform sampling strategies generally fail to provide consistent improvements across continual learning methods and settings. SBS is, to our knowledge, the first sampling method that consistently enhances different continual learning methods, making this a key contribution of our work.
>
> Some methods, like iCaRL [2], incorporate specific non-random sampling strategies (herding for iCaRL), but these are tightly coupled to the method itself and do not generalize well. For instance, herding does not work effectively with most competitive continual learning methods, while iCaRL is designed around herding and performs poorly with other sampling methods, including random sampling. However, for completeness, we will follow your suggestion and include additional results on continual learning methods with different sampling functions in the supplementary material and briefly discuss them in the main paper.
>
> **Max entropy and IPM performance**
>
> Max entropy (Max Ent) selects examples with the highest entropy in network logits. While a common baseline, it is rather a naive baseline, and it is known to perform poorly in practice, as also observed in our results.
>
> IPM [3] selects a diverse subset of examples from a dataset, and tries to maximize the information that this set has. The original IPM paper was not focused on continual learning, and showed advantages of this subset in other fields, such as active learning, representation learning, and GAN training. However, prior work [4] indicates that its effectiveness in continual learning is inconsistent, which aligns with our findings. A possible explanation is that IPM prioritizes "informative" examples, which often overlap with slower-to-learn examples. Since our analysis shows that slower-to-learn examples are more prone to forgetting, this selection bias may explain IPM’s weaker performance. Interestingly, IPM performs better in settings where learning is easier or buffer sizes are large (e.g., CIFAR-100-20 with a 10k buffer), which is consistent with this hypothesis.
>
> **Choice of hyper-parameters**
>
> As noted in our response to reviewer hXyj, the performance of $q=s=20\%$ is already present in multiple figures throughout the paper, including Figs. 4(a-b), 16(a-b), 17(a-b), 18(a-c), 19(a-c), 20(a-c), 21(a-c), and 24(a-c), showing its advantages over random sampling. However, to make this clearer for future readers, we will add a dedicated paragraph consolidating these results in the camera-ready version. Additionally, we will include SBS results with these fixed hyper-parameters in Table 1, demonstrating that they still significantly outperform random sampling without requiring additional tuning or additional computation. The exact numeric results for $q=s=20\%$ that will appear in Table 1 are also provided in our response to reviewer hXyj.
>
> --------------------------------------
>
> [1] Wang, Liyuan, et al. "A comprehensive survey of continual learning: Theory, method and application." (PAMI 2024)
>
> [2] Rebuffi, Sylvestre-Alvise, et al. "icarl: Incremental classifier and representation learning." (CVPR 2017)
>
> [3] Zaeemzadeh, Alireza, et al. "Iterative projection and matching: Finding structure-preserving representatives and its application to computer vision."  (CVPR 2019)
>
> [4] Brignac, Daniel, Niels Lobo, and Abhijit Mahalanobis. "Improving replay sample selection and storage for less forgetting in continual learning." (ICCV 2023)

---

### Decision · Program_Chairs · 2025-05-01

**Decision:**

Accept (poster)

**Comment:**

This paper uses the learning speed of examples to decide whether or not to store them in replay-based continual learning, finding improvements.

Reviewers agree that the method is well-motivated empirically. Reviewer 98Km asks for baselines beyond uniform sampling, and although I mostly agree with the authors' point that random uniform sampling performs very well (and often better than other methods), I think the paper would be stronger by including a couple of the better-performing sampling methods as baselines. I do note that the authors have Table 1 already.

All reviewers mention the issue of hyperparameter tuning, specifically the cost of the Rotnet task. The authors rebutted that q and s are usually set at the same values. I think this paper would be stronger if the authors also used another, cheaper, hyperparameter tuning strategy and showed that their method still performs well. This is especially important given that this paper is a more empirical and less theoretical work, and therefore it is important to show the benefits empirically.

It is extremely important to add class-incremental results. The authors promised to add these results in a future version but they are not there right now (or in the rebuttal text explicitly).